# Distributional Generalization: Characterizing Classifiers Beyond Test Error

## Abstract

We present a new set of empirical properties of interpolating classifiers, including neural networks, kernel machines and decision trees. Informally, the output distribution of an interpolating classifier matches the distribution of true labels, when conditioned on certain subgroups of the input space. For example, if we mislabel 30% of dogs as cats in the train set of CIFAR-10, then a ResNet trained to interpolation will in fact mislabel roughly 30% of dogs as cats on the *test set* as well, while leaving other classes unaffected. These behaviors are not captured by classical generalization, which would only consider the average error over the inputs, and not *where* these errors occur. We introduce and experimentally validate a formal conjecture that specifies the subgroups for which we expect this distributional closeness. Further, we show that these properties can be seen as a new form of generalization, which advances our understanding of the implicit bias of interpolating methods.

## 1 Introduction

In learning theory, when we study how well a classifier "generalizes", we usually consider a single metric – its test error [59]. However, there could be many different classifiers with the same test error that differ substantially in, say, the subgroups of inputs on which they make errors or in the features they use to attain this performance. Reducing classifiers to a single number misses these rich aspects of their behavior. In this work, we propose formally studying the entire *joint distribution* of classifier inputs and outputs. That is, the distribution $(x, f(x))$ for samples from the distribution $x \sim D$ for a classifier $f(x)$. This distribution reveals many structural properties of the classifier beyond test error (such as *where* the errors occur). In fact, we discover new behaviors of modern classifiers that can only be understood in this framework. As an example, consider the following experiment (Figure 1).

**Experiment 1.** *Consider a binary classification version of CIFAR-10, where CIFAR-10 images $x$ have binary labels* `Animal`/`Object`. *Take 50K samples from this distribution as a train set, but apply the following label noise: flip the label of cats to* `Object` *with probability 30%. Now train a WideResNet $f$ to 0 train error on this train set. How does the trained classifier behave on test samples? Options below:*

**(1)** The test error is low across all classes, since there is only 3% overall label noise in the train set.

**(2)** Test error is "spread" across the animal class. After all, the classifier is not explicitly told what a cat or a dog is, just that they are all animals.

**(3)** The classifier misclassifies roughly 30% of test cats as "objects", but all other animals are largely unaffected.

The reality is closest to option (3) as shown in Figure 1. The left panel shows the joint density of train inputs $x$ with train labels `Object`/`Animal`. Since the classifier is interpolating, the classifier

Submitted to 35th Conference on Neural Information Processing Systems (NeurIPS 2021). Do not distribute.

outputs on the train set are identical to the left panel. The right panel shows the *classifier predictions* $f(x)$ on *test inputs* $x$.

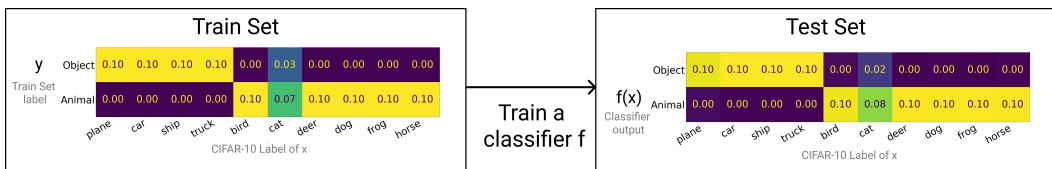

Figure 1: The setup and result of Experiment 1. The CIFAR-10 train set is labeled as either Animals or Objects, with label noise affecting only cats. A WideResNet-28-10 is then trained to 0 train error on this train set, and evaluated on the test set. Full experimental details in Appendix C.2

There are several notable things about this experiment. First, the error is *localized* to cats in the test set as it was in the train set, even though no explicit cat labels were provided. The interpolating model is thus sensitive to subgroup-structures in the distribution. Second, the *amount* of error on the cat class is close to the noise applied on the train set. Thus, the behavior of the classifier on the train set *generalizes* to the test set in a stronger sense than just average error. Specifically, when *conditioned on a subgroup* (cat), the *distribution* of the true labels is close to that of the classifier outputs. Third, this is not the behavior of the Bayes-optimal classifier, which would always output the maximum-likelihood label instead of reproducing the noise in the distribution. The network is thus behaving poorly from the perspective of Bayes-optimality, but behaving well in a certain distributional sense (which we will formalize soon).

Now, consider a seemingly unrelated experimental observation. Take an AlexNet trained on ImageNet, a 1000-way classification problem with 116 varieties of dogs. AlexNet only achieves 56.5% test accuracy on ImageNet. However, it at least classifies most dogs as *some* variety of dog (with 98.4% accuracy), though it may mistake the exact breed. In this work, we show that both of these experiments are examples of the same underlying phenomenon. We empirically show that for an interpolating classifier, its classification outputs are close in distribution to the true labels — even when conditioned on many subsets of the domain. For example, in Figure 1, the distribution of $p(f(x)|x = \text{cat})$ is close to the true label distribution of $p(y|x = \text{cat})$. We propose a formal conjecture (Feature Calibration), that predicts which subgroups of the domain can be conditioned on for the above distributional closeness to hold.

These experimental behaviors could not have been captured solely by looking at average test error, as is done in the classical theory of generalization. In fact, they are special cases of a new kind of generalization, which we call "Distributional Generalization".

## 1.1 Distributional Generalization

Informally, Distributional Generalization states that the outputs of classifiers $f$ on their train sets and test sets are close *as distributions* (as opposed to close in just error). That is, the following joint distributions[1] are close:

$$(x, f(x))_{x \sim \text{TestSet}} \approx (x, f(x))_{x \sim \text{TrainSet}} \tag{1}$$

The remainder of this paper is devoted to making the above statement precise, and empirically checking its validity on real-world tasks. Specifically, we want to formally define the notion of approximation ($\approx$), and understand how it depends on the problem parameters (the type of classifier, number of train samples, etc). We focus primarily on interpolating methods, where we formalize Equation (1) through our Feature Calibration Conjecture.

## 1.2 Our Contributions and Organization

In this work, we discover new empirical properties of interpolating classifiers, which are not captured in the classical framework of generalization. We then propose formal conjectures to characterize these behaviors.

---

[1]These distributions also include the randomness in sampling the train and test sets, and in training the classifier, as we define more precisely in Section 3.

- In Section 3, we introduce a formal "Feature Calibration" conjecture, which unifies our experimental observations. Roughly, Feature Calibration says that the outputs of classifiers match the statistics of their training distribution when conditioned on certain subgroups.

- In Section 4, we experimentally stress test our Feature Calibration conjecture across various settings in machine learning, including neural networks, kernel machines, and decision trees. This highlights the universality of our results across machine learning.

- In Section 5, we relate our results to classical generalization, by defining a new notion of Distributional Generalization which subsumes both classical generalization and our new conjectures.

- Finally, in Section 5.2 we informally discuss how Distributional Generalization can be applied even for non-interpolating methods.

Our results, thus, extend our understanding of the *implicit bias* of interpolating methods, and introduce a new type of generalization exhibited across many methods in machine learning.

## 1.3 Related Work and Significance

Our work has connections to, and implications for many existing research programs in deep learning.

**Implicit Bias and Overparameterization.** There has been a long line of recent work towards understanding overparameterized and interpolating methods, since these pose challenges for classical theories of generalization (e.g. Belkin et al. [8, 9, 10], Breiman [11], Gunasekar et al. [25], Liang and Rakhlin [36], Nakkiran et al. [43], Schapire et al. [58], Soudry et al. [62], Zhang et al. [71]). The "implicit bias" program here aims to answer: *Among all models with 0 train error, which model is actually produced by SGD?* Most existing work seeks to characterize the exact implicit bias of models under certain (sometimes strong) assumptions on the model, training method or the data distribution. In contrast, our conjecture applies across many different interpolating models (from neural nets to decision trees) as they would be used in practice, and thus form a sort of "universal implicit bias" of these methods. Moreover, our results place constraints on potential future theories of implicit bias, and guide us towards theories that better capture practice.

**Benign Overfitting.** Most prior works on interpolating classifiers attempt to explain why training to interpolation "does not harm" the the model. This has been dubbed "benign overfitting" [7] and "harmless interpolation" [40], reflecting the widely-held belief that interpolation does not harm the decision boundary of classifiers. In contrast, we find that interpolation actually does "harm" classifiers, in predictable ways: fitting the label noise on the train set causes similar noise to be reproduced at test time. Our results thus indicate that interpolation can significantly affect the decision boundary of classifiers, and should not be considered a purely "benign" effect.

**Classical Generalization and Scaling Limits.** Our framework of Distributional Generalization is insightful even to study classical generalization, since it reveals much more about models than just their test error. For example, statistical learning theory attempts to understand if and when models will asymptotically converge to Bayes optimal classifiers, in the limit of large data ("asymptotic consistency" [59, 65]). In deep learning, there are at least two distinct ways to scale model and data to infinity together: the *underparameterized* scaling limit, where data-size $\gg$ model-size always, and the *overparameterized* scaling limit, where data-size $\ll$ model-size always. The underparameterized scaling limit is well-understood: when data is essentially infinite, neural networks will converge to the Bayes-optimal classifier (provided the model-size is large enough, and the optimization is run for long enough, with enough noise to escape local minima). On the other hand, our work suggests that in the *overparameterized* scaling limit, models will *not* converge to the Bayes-optimal classifier. Specifically, our Feature Calibration Conjecture implies that in the limit of large data, interpolating models will approach a *sampler* from the distribution. That is, the limiting model $f$ will be such that the output $f(x)$ is a sample from $p(y|x)$, as opposed to the Bayes-optimal $f^*(x) = \text{argmax}_y p(y|x)$. This claim— that overparameterized models do not converge to Bayes-optimal classifiers— is unique to our work as far as we know, and highlights the broad implications of our results.

**Locality and Manifold Learning.** Our intuition for the behaviors in this work is that they arise due to some form of "locality" of the trained classifiers, in an appropriate embedding space. For example, the behavior observed in Experiment 1 would be consistent with that of a 1-Nearest-Neighbor classifier in a embedding that separates the CIFAR-10 classes well. This intuition that classifiers learn good

embeddings is present in various forms in the literature, for example: the so-called called "manifold hypothesis," that natural data lie on a low-dimensional manifold [44, 61], as well as works on local stiffness of the loss landscape [19], and works showing that overparameterized neural networks can learn hidden low-dimensional structure in high-dimensional settings [6, 15, 21]. It is open to more formally understand connections between our work and the above.

**Other Related Works.** Our conjectures also describe neural networks under label noise, which has been empirically and theoretically studied in the past [9, 14, 45, 54, 63, 71, 72], though not formally characterized. A full discussion of related works is in Appendix A.

## 2 Preliminaries

**Notation.** We consider joint distributions $\mathcal{D}$ on $x \in \mathcal{X}$ and discrete $y \in \mathcal{Y} = [k]$. Let $S = \{(x_i, y_i)\}_{i=1}^n \sim \mathcal{D}^n$ denote a train set of $n$ iid samples from $\mathcal{D}$. Let $\mathcal{A}$ denote the training procedure (including architecture and training algorithm for neural networks), and let $f \leftarrow \mathrm{Train}_\mathcal{A}(S)$ denote training a classifier $f$ on train-set $S$ using procedure $\mathcal{A}$. We consider classifiers which output hard decisions $f : \mathcal{X} \to \mathcal{Y}$. Let $\mathrm{NN}_S(x) = x_i$ denote the nearest-neighbor to $x$ in train-set $S$, with respect to a distance metric $d$. Our theorems will apply to any distance metric, and so we leave this unspecified. Let $\mathrm{NN}_S^{(y)}(x)$ denote the nearest-neighbor estimator itself, that is, $\mathrm{NN}_S^{(y)}(x) := y_i$ where $x_i = \mathrm{NN}_S(x)$.

**Experimental Setup.** Briefly, we train all classifiers to interpolation (to 0 train error). Neural networks (MLPs and ResNets [29]) are trained with SGD. Interpolating decision trees are trained using the growth rule from Random Forests [12]. For kernel classification, we consider kernel regression on one-hot labels and kernel SVM, with small or 0 of regularization (which is often optimal [60]). Full experimental details are provided in Appendix B.

**Distributional Closeness.** We consider the following notion of closeness for two probability distributions: For two distributions $P, Q$ over $\mathcal{X} \times \mathcal{Y}$, let a "test" (or "distinguisher") be a function $T : \mathcal{X} \times \mathcal{Y} \to [0, 1]$ which accepts a sample from either distribution, and is intended to classify the sample as either from distribution $P$ or $Q$. For any set $\mathcal{C} \subseteq \{T : \mathcal{X} \times \mathcal{Y} \to [0, 1]\}$ of tests, we say distributions $P$ and $Q$ are "$\varepsilon$-indistinguishable up to $\mathcal{C}$-tests" if they are close with respect to all tests in class $\mathcal{C}$. That is,

$$P \approx_\varepsilon^\mathcal{C} Q \iff \sup_{T \in \mathcal{C}} \left| \mathop{\mathbb{E}}_{(x,y) \sim P}[T(x,y)] - \mathop{\mathbb{E}}_{(x,y) \sim Q}[T(x,y)] \right| \leq \varepsilon \tag{2}$$

Total-Variation distance is equivalent to closeness in all tests, i.e. $\mathcal{C} = \{T : \mathcal{X} \times \mathcal{Y} \to [0, 1]\}$, but we consider closeness for restricted families of tests $\mathcal{C}$. $P \approx_\varepsilon Q$ denotes $\varepsilon$-closeness in TV-distance.

## 3 Feature Calibration Conjecture

### 3.1 Distributions of Interest

We first define three key distributions that we will use in stating our formal conjecture. For a given data distribution $\mathcal{D}$ over $\mathcal{X} \times \mathcal{Y}$ and training procedure $\mathrm{Train}_\mathcal{A}$, we consider the following three distributions over $\mathcal{X} \times \mathcal{Y}$:

1. **Source $\mathcal{D}$:** $(x, y)$ where $x, y \sim \mathcal{D}$.
2. **Train $\mathcal{D}_{\mathrm{tr}}$:** $(x_{\mathrm{tr}}, f(x_{\mathrm{tr}}))$ where $S \sim \mathcal{D}^n$, $f \leftarrow \mathrm{Train}_\mathcal{A}(S)$, $(x_{\mathrm{tr}}, y_{\mathrm{tr}}) \sim S$
3. **Test $\mathcal{D}_{\mathrm{te}}$:** $(x, f(x))$ where $S \sim \mathcal{D}^n$, $f \leftarrow \mathrm{Train}_\mathcal{A}(S)$, $x, y \sim \mathcal{D}$

The source distribution $\mathcal{D}$ is simply the original distribution. To sample once from the **Train Distribution** $\mathcal{D}_{\mathrm{tr}}$, we first sample a train set $S \sim \mathcal{D}^n$, train a classifier $f$ on it, then output $(x_{\mathrm{tr}}, f(x_{\mathrm{tr}}))$ for a random *train point* $x_{\mathrm{tr}} \in S$. That is, $\mathcal{D}_{\mathrm{tr}}$ is the distribution of input and outputs of a trained classifier $f$ on its train set. To sample once from the **Test Distribution** $\mathcal{D}_{\mathrm{te}}$, we do this same procedure, but output $(x, f(x))$ for a random *test point* $x$. That is, the $\mathcal{D}_{\mathrm{te}}$ is the distribution of input and outputs of a trained classifier $f$ at test time. The only difference between the Train Distribution and

Test Distribution is that the point $x$ is sampled from the train set or the test set, respectively.[2] For interpolating classifiers, $f(x_{\mathrm{tr}}) = y_{\mathrm{tr}}$ on the train set, and so the Source and Train distributions are equivalent: $\mathcal{D} \equiv \mathcal{D}_{\mathrm{tr}}$. (Note that these definitions, crucially, involve randomness from sampling the train set, training the classifier, and sampling a test point).

## 3.2 Feature Calibration

We now formally describe the Feature Calibration Conjecture. At a high level, we argue that the distributions $\mathcal{D}_{\mathrm{te}}$ and $\mathcal{D}$ are statistically close for interpolating classifiers if we first "coarsen" the domain of $x$ by some partition $L : \mathcal{X} \to [M]$ in to $M$ parts. That is, for certain partitions $L$, the following distributions are statistically close:

$$(L(x), f(x))_{x \sim \mathcal{D}} \approx_\varepsilon (L(x), y)_{x \sim \mathcal{D}}$$

We think of $L$ as defining subgroups over the domain— for example, $L(x) \in \{\text{dog, cat, horse}\dots\}$. Then, the above statistical closeness is essentially equivalent to requiring that for all subgroups $\ell \in [M]$, the conditional distribution of classifier output on the subgroup—$p(f(x)|L(x) = \ell)$ — is close to the true conditional distribution: $p(y|L(x) = \ell)$.

The crux of our conjecture lies in defining exactly which subgroups $L$ satisfy this distributional closeness, and quantifying the $\varepsilon$ approximation. This is subtle, since it must depend on almost all parameters of the problem. For example, consider a modification to Experiment 1, where we use a fully-connected network (MLP) instead of a ResNet. An MLP cannot properly distinguish cats even when it is actually provided the real CIFAR-10 labels, and so (informally) it has no hope of behaving differently on cats in the setting of Experiment 1, where the cats are not labeled explicitly (See Figure C.2 for results with MLPs). Similarly, if we train the ResNet with very few samples from the distribution, the network will be unable to recognize cats. Thus, the allowable partitions must depend on the classifier family and the training method, including the number of samples.

We conjecture that allowable partitions are those which can themselves be learnt to good test performance with an identical training procedure, but trained with the labels of the partition $L$ instead of $y$. To formalize this, we define a *distinguishable feature*: a partition of the domain $\mathcal{X}$ that is learnable for a given training procedure. Thus, in Experiment 1, the partition into CIFAR-10 classes would be a distinguishable feature for ResNets (trained with SGD with 50K or more samples), but not for MLPs. The definition below depends on the training procedure $\mathcal{A}$, the data distribution $\mathcal{D}$, number of train samples $n$, and an approximation parameter $\varepsilon$ (which we think of as $\varepsilon \approx 0$).

**Definition 1** (($\varepsilon, \mathcal{A}, \mathcal{D}, n$)-Distinguishable Feature). *For a distribution $\mathcal{D}$ over $\mathcal{X} \times \mathcal{Y}$, number of samples $n$, training procedure $\mathcal{A}$, and small $\varepsilon \geq 0$, an ($\varepsilon, \mathcal{A}, \mathcal{D}, n$)-distinguishable feature is a partition $L : \mathcal{X} \to [M]$ of the domain $\mathcal{X}$ into $M$ parts, such that training a model using $\mathcal{A}$ on $n$ samples labeled by $L$ works to classify $L$ with high test accuracy. Precisely, $L$ is a ($\varepsilon, \mathcal{A}, \mathcal{D}, n$)-distinguishable feature if:*

$$\Pr_{\substack{S=\{(x_i, L(x_i)\}_{x_1,\dots,x_n \sim \mathcal{D}} \\ f \leftarrow \mathrm{Train}_{\mathcal{A}}(S); \ x \sim \mathcal{D}}} [f(x) = L(x)] \geq 1 - \varepsilon$$

This definition depends only on the marginal distribution of $\mathcal{D}$ on $x$, and not on the label distribution $p_{\mathcal{D}}(y|x)$. To recap, this definition is meant to capture a labeling of the domain $\mathcal{X}$ that is learnable for a given training procedure $\mathcal{A}$. It must depend on the architecture used by $\mathcal{A}$ and number of samples $n$, since more powerful classifiers can distinguish more features. Note that there could be many distinguishable features for a given setting ($\varepsilon, \mathcal{A}, \mathcal{D}, n$) — including features not implied by the class label such as the presence of grass in a CIFAR-10 image. Our main conjecture follows.

**Conjecture 1** (Feature Calibration). *For all natural distributions $\mathcal{D}$, number of samples $n$, interpolating training procedures $\mathcal{A}$, and $\varepsilon \geq 0$, the following distributions are statistically close for all ($\varepsilon, \mathcal{A}, \mathcal{D}, n$)-distinguishable features $L$:*

$$\underset{f \leftarrow \mathrm{Train}_{\mathcal{A}}(\mathcal{D}^n); \ x, y \sim \mathcal{D}}{(L(x), f(x))} \quad \approx_\varepsilon \quad \underset{x, y \sim \mathcal{D}}{(L(x), y)} \tag{3}$$

*or equivalently:*

$$\underset{x, \widehat{y} \sim \mathcal{D}_{\mathrm{te}}}{(L(x), \widehat{y})} \quad \approx_\varepsilon \quad \underset{x, y \sim \mathcal{D}}{(L(x), y)} \tag{4}$$

---

[2]Technically, these definitions require training a fresh classifier for each sample, using independent train sets. For practical reasons most of our experiments train a single classifier $f$ and evaluate it on the entire train/test set.

This claims that the TV distance between the LHS and RHS of Equation (4) is at most $\varepsilon$, where $\varepsilon$ is the error of the distinguishable feature (in Definition 1). We claim that this holds *for all* distinguishable features $L$ "automatically" – we simply train a classifier, without specifying any particular partition. The formal statements of Definition 1 and Conjecture 1 may seem somewhat arbitrary, involving many quantifiers over $(\varepsilon, \mathcal{A}, \mathcal{D}, n)$. However, we believe these statements are natural: In addition to extensive experimental evidence in Section 4, we also prove that Conjecture 1 is formally true as stated for 1-Nearest-Neighbor classifiers in Theorem 1.

### 3.3 Feature Calibration for 1-Nearest-Neighbors

Here we prove that the 1-Nearest-Neighbor classifier formally satisfies Conjecture 1, under mild assumptions. We view this theorem as support for our (somewhat involved) formalism of Conjecture 1. Indeed, without Theorem 1 below, it is unclear if our statement of Conjecture 1 can ever be satisfied by any classifier, or if it is simply too strong to be true. This theorem applies generically to a wide class of distributions; the only assumption is a weak regularity condition: sampling the nearest-neighbor train point to a random test point should yield (close to) a uniformly random test point.

**Theorem 1.** *Let $\mathcal{D}$ be a distribution over $\mathcal{X} \times \mathcal{Y}$, and let $n \in \mathbb{N}$ be the number of train samples. Assume the following regularity condition holds: Sampling the nearest-neighbor train point to a random test point yields (close to) a uniformly random test point. That is, suppose that for some small $\delta \geq 0$, the distributions:* $\{\mathrm{NN}_S(x)\}_{\substack{S \sim \mathcal{D}^n \\ x \sim \mathcal{D}}} \approx_\delta \{x\}_{x \sim \mathcal{D}}$. *Then, Conjecture 1 holds. That is, for all $(\varepsilon, \mathrm{NN}, \mathcal{D}, n)$-distinguishable partitions $L$, the following distributions are statistically close:*

$$\{(y, L(x))\}_{x,y \sim \mathcal{D}} \quad \approx_{\varepsilon+\delta} \quad \{(\mathrm{NN}_S^{(y)}(x), L(x)\}_{\substack{S \sim \mathcal{D}^n \\ x,y \sim \mathcal{D}}} \tag{5}$$

The proof of Theorem 1 is straightforward, and provided in Appendix D – but this strong property of nearest-neighbors was not know before, to our knowledge.

### 3.4 Limitations: Natural Distributions

Technically, Conjecture 1 is not fully specified, since it does not specify exactly which classifiers or distributions obey the conjecture. We do not claim that *all* classifiers and distributions satisfy our conjectures. Nevertheless, we claim our conjectures hold in all "natural" settings, which informally means settings with real data and classifiers that are actually used in practice. The problem of understanding what separates "natural distributions" from artificial ones is not unique to our work, and lies at the heart of deep learning theory. Many theoretical works handle this by considering simplified distributional assumptions (e.g. smoothness, well-separatedness, gaussianity), which are mathematically tractable, but untested in practice [2, 4, 35]. In contrast, we do not make untestable mathematical assumptions. This benefit of realism comes at the cost of mathematical formalism. We hope that as the theory of deep learning evolves, we will better understand how to formalize the notion of "natural" in our conjectures.

## 4 Experiments: Feature Calibration

We now give empirical evidence for our conjecture in a variety of settings in machine learning, including neural networks, kernel machines, and decision trees. In each experiment, we consider a feature that is (verifiably) distinguishable, and then test our Feature Calibration conjecture for this feature. Each of the experimental settings below highlights a different aspect of interpolating classifiers, which may be of independent interest. Selected experiments are summarized here, with full details and further experiments in Appendix C.

**Constant Partition:** Consider the trivially-distinguishable *constant* feature: $L(x) = 0$ everywhere. For this feature, Conjecture 1 reduces to the statement that the marginal distribution of class labels for any interpolating classifier is close to the true marginals $p(y)$. To test this, we construct a variant of CIFAR-10 with class-imbalance and train classifiers with varying levels of test errors to interpolation on it. As shown in Figure 2B, the marginals of the classifier outputs are close to the true marginals, even for a classifier that only achieves 37% test error.

**Coarse Partition:** Consider AlexNet trained on ILSVRC-2012 ImageNet [56], a 1000-class image classification problem with 116 varieties of dogs. The network achieves only 56.5% accuracy

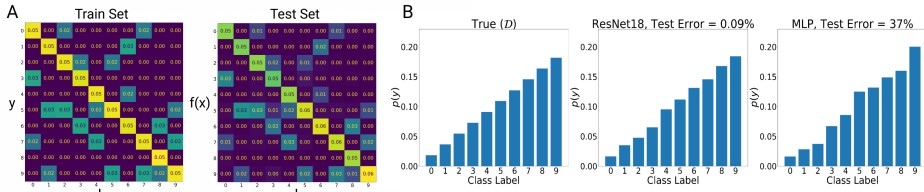

Figure 2: **Feature Calibration.** **(A)** Random confusion matrix on CIFAR-10, with a WideResNet28-10 trained to interpolation. Left: Joint density of labels $y$ and original class $L$ on the train set. Right: Joint density of classifier predictions $f(x)$ and original class $L$ on the test set. These two joint densities are close, as predicted by Conjecture 1. **(B)** Constant partition: The CIFAR-10 train set is class-rebalanced according to the left panel distribution. The center and right panels show that both ResNets and MLPs have the correct marginal distribution of outputs, even though the MLP has high test error.

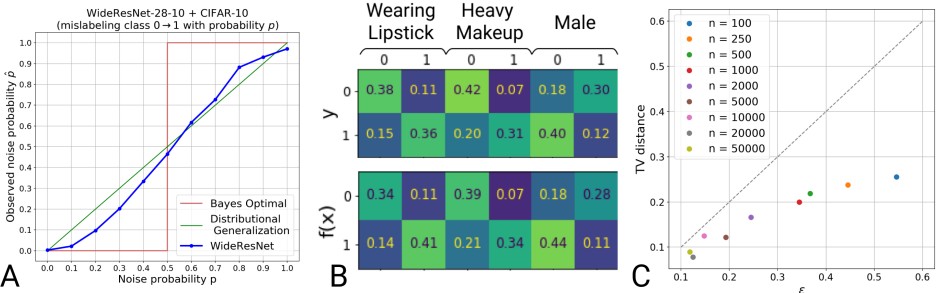

Figure 3: **Feature Calibration.** **(A)** CIFAR-10 with $p$ fraction of class $0 \to 1$ mislabeled on the train set. Plotting observed noise on classifier outputs vs. applied noise on the train set. **(B)** Multiple feature calibration on CelebA. **(C)** TV-distance between $(L(x), f(x))$ and $(L(x), y)$ for a variant of Experiment 1 with error on the distinguishable partitions ($\varepsilon$). The error was changed by changing the number of samples $n$.

on the test set. But it will at least classify most dogs as dogs (with 98.4% accuracy), making $L(x) \in \{\text{dog, not-dog}\}$ a distinguishable feature. Moreover, as predicted by Conjecture 1, the network is *calibrated* with respect to dogs: 22.4% of all dogs in ImageNet are Terriers, and indeed the network classifies 20.9% of all dogs as Terriers (though it has 9% error on which specific dogs it classifies as Terriers). See Appendix Table 2 for details, and related experiments on ResNets and kernels in Appendix C.

**Class Partition:** We now consider settings where the class labels are themselves distinguishable features (eg: CIFAR-10 classes are distinguishable by ResNets). Here our conjecture predicts the behavior of interpolating classifiers under structured label noise. As an example, we generate a random spare confusion matrix and apply this to the labels of CIFAR-10 as shown in Figure 2A. We find that a WideResNet trained to interpolation outputs the same confusion matrix on the test set as well (Figure 2B). Now, to test that this phenomenon is indeed robust to the level of noise, we mislabel class $0 \to 1$ with probability $p$ in the CIFAR-10 train set for varying levels of $p$. We then observe $\hat{p}$, the fraction of samples mislabeled by this network from $0 \to 1$ in the test set (Figure 3A shows $p$ versus $\hat{p}$). The Bayes optimal classifier for this distribution behaves as a step function (in red), and a classifier that obeys Conjecture 1 exactly would follow the diagonal (in green). The actual experiment (in blue) is close to the behavior predicted by Conjecture 1. This experiment shows a contrast with classical learning theory. While most existing theory focuses on whether classifiers converge to the Bayes optimal solution, we show that interpolating classifiers behave "optimally" in a different sense: they match the distribution of their train set. We discuss this further in Section 5. See Appendix C.4 for more experiments, including other classifiers such as Decisions Trees.

**Multiple features:** Conjecture 1 states that the network should be automatically calibrated for all distinguishable features, without any explicit labels for them. To do this, we use the CelebA dataset [37], containing images with many binary attributes per image. ("male", "blond hair", etc).

We train a ResNet-50 to classify one of the hard attributes (accuracy 80%) and confirm that the Feature Calibration holds for all the other attributes (Figure 3) that are themselves distinguishable.

**Quantitative predictions:** We now test the quantitative predictions made by Conjecture 1. This conjecture states that the TV-distance between the joint distributions $(L(x), f(x))$ and $(L(x), y)$ is at most $\varepsilon$, where $\varepsilon$ is the error of the training procedure in learning $L$ (see Definition 1). To test this, we consider binary task similar to Experiment 1 where (Ship, Plane) are labeled as class 0 and (Cat, Dog) are labeled as class 1, with $p = 0.3$ fraction of cats mislabeled to class 0. Then, we train a convolutional network to interpolation on this task. To vary the error $\varepsilon$ on these distinguishable features systematically, we train networks with varying number of train samples. Networks with fewer samples have larger $\varepsilon$ since they are worse at classifying the distinguishable features of (Ship,Plane,Cat,Dog). Then, we use the same setup to train networks on the binary task and measure the TV-distance between $(L(x), f(x))$ and $(L(x), y)$ in this task. The results are shown in Figure 3C. As predicted, the TV distance on the binary task is upper bounded by $\varepsilon$ error on the 4-way classification task.

## 5 Distributional Generalization

In order to relate our results to the classical theory of generalization, we now propose a formal notion of "Distributional Generalization", which subsumes both Feature Calibration and classical generalization. In fact, we will also give preliminary evidence that this new notion can apply even for non-interpolating methods, unlike Feature Calibration.

A trained model $f$ obeys classical generalization (with respect to test error) if its error on the train set is close to its error on the test distribution. We first rewrite this using our definitions below.

**Classical Generalization (informal):** *Let $f$ be a trained classifier. Then $f$ generalizes if:*

$$\underset{\substack{x \sim TrainSet \\ \widehat{y} \leftarrow f(x)}}{\mathbb{E}} [\mathbb{1}\{\widehat{y} \neq y(x)\}] \approx \underset{\substack{x \sim TestSet \\ \widehat{y} \leftarrow f(x)}}{\mathbb{E}} [\mathbb{1}\{\widehat{y} \neq y(x)\}] \tag{6}$$

Above, $y(x)$ is the true class of $x$ and $\widehat{y}$ is the predicted class. The LHS of Equation 6 is the train error of $f$, and the RHS is the test error. Using our definitions of $\mathcal{D}_{\text{tr}}, \mathcal{D}_{\text{te}}$ from Section 3.1, and defining $T_{\text{err}}(x, \widehat{y}) := \mathbb{1}\{\widehat{y} \neq y(x)\}$, we can write Equation 6 equivalently:

$$\underset{x, \widehat{y} \sim \mathcal{D}_{\text{tr}}}{\mathbb{E}} [T_{\text{err}}(x, \widehat{y})] \approx \underset{x, \widehat{y} \sim \mathcal{D}_{\text{te}}}{\mathbb{E}} [T_{\text{err}}(x, \widehat{y})] \tag{7}$$

That is, classical generalization states that a certain function ($T_{\text{err}}$) has similar expectations on both the Train Distribution $\mathcal{D}_{\text{tr}}$ and Test Distribution $\mathcal{D}_{\text{te}}$. We can now introduce Distributional Generalization, which is a property of trained classifiers. It is parameterized by a set of bounded functions ("tests"): $\mathcal{T} \subseteq \{T : \mathcal{X} \times \mathcal{Y} \rightarrow [0, 1]\}$.

**Distributional Generalization:** *Let $f$ be a trained classifier. Then $f$ satisfies Distributional Generalization with respect to tests $\mathcal{T}$ if:*

$$\forall T \in \mathcal{T} : \underset{x, \widehat{y} \sim \mathcal{D}_{\text{tr}}}{\mathbb{E}} [T(x, \widehat{y})] \approx \underset{x, \widehat{y} \sim \mathcal{D}_{\text{te}}}{\mathbb{E}} [T(x, \widehat{y})] \tag{8}$$

This states that the train and test distribution have similar expectations for *all* functions in the family $\mathcal{T}$, which we can write as: $\mathcal{D}_{\text{tr}} \approx^{\mathcal{T}} \mathcal{D}_{\text{te}}$. For the singleton set $\mathcal{T} = \{T_{\text{err}}\}$, this is equivalent to classical generalization, but it may hold for much larger sets $\mathcal{T}$. This definition of Distributional Generalization, like the definition of classical generalization, is just defining an object— not stating when or how it is satisfied. Feature Calibration turns this into a concrete conjecture.

### 5.1 Feature Calibration as Distributional Generalization

We can write our Feature Calibration Conjecture as a special case of Distributional Generalization, for a certain family of tests $\mathcal{T}$. Informally, for a given setting, the family $\mathcal{T}$ is all tests which take input $(x, y)$, but only depend on $x$ via a *distinguishable feature* (Definition 1). For example, a test of the form $T(x, y) = g(L(x), y)$ where $L$ is a distinguishable feature, and $g$ is arbitrary. Formally, for a given problem setting, suppose $\mathcal{L}$ is the set of $(\varepsilon, \mathcal{A}, \mathcal{D}, n)$-distinguishable features. Then Conjecture 1 states that $\forall L \in \mathcal{L} : (L(x), f(x)) \approx_{\varepsilon} (L(x), y)$. This is equivalent to the statement

$$\mathcal{D}_{\text{te}} \approx_{\varepsilon}^{\mathcal{T}} \mathcal{D} \tag{9}$$

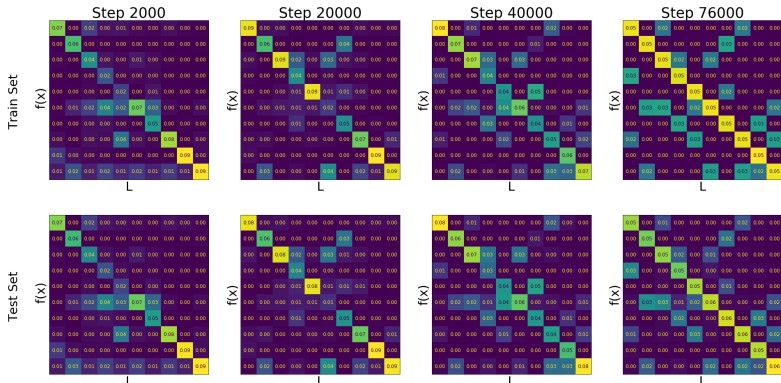

Figure 4: **Distributional Generalization for WideResNet on CIFAR-10.** The confusion matrices on the train set (top row) and test set (bottom row) remain close throughout training.

where $\mathcal{T}$ is the set of functions $\mathcal{T} := \{T : T(x,y) = g(L(x), y), \ \ L \in \mathcal{L}, \ \ g : \mathcal{X} \times \mathcal{Y} \to [0,1]\}$. For interpolating classifiers, we have $\mathcal{D} \equiv \mathcal{D}_{\mathrm{tr}}$, and so Equation (9) is equivalent to $\mathcal{D}_{\mathrm{te}} \approx_{\varepsilon}^{\mathcal{T}} \mathcal{D}_{\mathrm{tr}}$, which is a statement of Distributional Generalization. Since any classifier family will contain a large number of distinguishable features, the set $\mathcal{L}$ may be very large. Hence, the distributions $\mathcal{D}_{\mathrm{tr}}$ and $\mathcal{D}_{\mathrm{te}}$ can be thought of as being close *as distributions*.

## 5.2 Beyond Interpolating Methods

The previous sections have focused on *interpolating* classifiers, which fit their train sets exactly. Here we informally discuss how to extend our results beyond interpolating methods. The discussion in this section is not as precise as in previous sections, and is only meant to suggest that our abstraction of Distributional Generalization can be useful in other settings.

For non-interpolating classifiers, we may still expect that they behave similarly on their test and train sets – that is, $\mathcal{D}_{\mathrm{te}} \approx^{\mathcal{T}} \mathcal{D}_{\mathrm{tr}}$ for some family of tests $\mathcal{T}$. For example, the following is a possible generalization of Feature Calibration to non-interpolating methods.

**Conjecture 2** (Generalized Feature Calibration, informal)**.** *For trained classifiers $f$, the following distributions are statistically close for many partitions $L$ of the domain:*

$$\underset{x,\widehat{y} \sim \mathcal{D}_{\mathrm{te}}}{(L(x), \widehat{y})} \quad \approx \quad \underset{x,\widehat{y} \sim \mathcal{D}_{\mathrm{tr}}}{(L(x), \widehat{y})} \tag{10}$$

We leave unspecified the exact set of partitions $L$ for which this holds, since we do not yet understand the appropriate notion of "distinguishable feature" in this setting. However, we give experimental evidence suggesting some refinement of Conjecture 2 is true. In Figure 4, we apply label noise from a random sparse confusion to the CIFAR-10 train set. We then train a single WideResNet28-10, and measure its predictions on the train and test sets over increasing train time (SGD steps). The top row shows the confusion matrix of predictions $f(x)$ vs true labels $L(x)$ on the train set, and the bottom row shows the corresponding confusion matrix on the test set. As the network is trained for longer, it fits more of the noise on the train set, and this noise is mirrored almost identically on the test set. Full experimental details, and an analogous experiment for kernels, are given in Appendix B.

## 6 Conclusion

This work initiates the study of a new kind of generalization— Distributional Generalization— which considers the entire input-output behavior of classifiers, instead of just their test error. We presented both new empirical behaviors, and new formal conjectures which characterize these behaviors. Roughly, our conjecture states that the outputs of classifiers on the test set are "close in distribution" to their outputs on the train set. These results build a deeper understanding of models used in practice, and we hope our results inspire further work on distributional generalization in machine learning.

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
