# A Full Related Work

Our work is inspired by the broader study of interpolating and overparameterized methods in machine learning; a partial list of works in this theme includes Advani and Saxe [1], Allen-Zhu et al. [3], Arora et al. [4], Bartlett et al. [7], Belkin et al. [8, 9, 10], Breiman [11], Chizat and Bach [15], Dziugaite and Roy [17], Geiger et al. [20], Gerace et al. [21], Ghorbani et al. [22], Goldt et al. [24], Hastie et al. [28], Liang and Rakhlin [36], Mei and Montanari [38], Muthukumar et al. [40], Nakkiran et al. [43], Neal et al. [46], Neyshabur et al. [47], Schapire et al. [58], Zhang et al. [71].

**Interpolating Methods.** Many of the best-performing techniques on high-dimensional tasks are interpolating methods, which fit their train samples to 0 train error. This includes neural-networks and kernels on images [29, 60], and random forests on tabular data [18]. Interpolating methods have been extensively studied both recently and in the past, since we do not theoretically understand their practical success [8–11, 28, 36, 38, 43, 57, 58, 71]. In particular, much of the classical work in statistical learning theory (uniform convergence, VC-dimension, Rademacher complexity, regularization, stability) fails to explain the success of interpolating methods [8, 9, 42, 71]. The few techniques which do apply to interpolating methods (e.g. margin theory [58]) remain vacuous on modern neural-networks and kernels.

**Decision Trees.** In a similar vein to our work, Olson and Wyner [49], Wyner et al. [67] investigate decision trees, and show that random forests are equivalent to a Nadaraya–Watson smoother [41, 66] with a certain smoothing kernel. Decision trees [13] are often intuitively thought of as "adaptive nearest-neighbors," since they are explicitly a spatial-partitioning method [27]. Thus, it may not be surprising that decision trees behave similarly to 1-Nearest-Neighbors. Olson and Wyner [49], Wyner et al. [67] took steps towards characterizing and understanding this behavior – in particular, Olson and Wyner [49] defines an equivalent smoothing kernel corresponding to a random forest, and empirically investigates the quality of the conditional density estimate. Our work presents a formal characterization of the quality of this conditional density estimate (Conjecture 1), which is a novel characterization even for decision trees, as far as we know.

**Kernel Smoothing.** The term kernel regression is sometimes used in the literature to refer to kernel *smoothers*, such as the Nadaraya–Watson kernel smoother [41, 66]. But in this work we use the term "kernel regression" to refer only to regression in a Reproducing Kernel Hilbert Space, as described in the experimental details.

**Label Noise.** Our conjectures also describe the behavior of neural networks under label noise, which has been empirically and theoretically studied in the past, though not formally characterized before [9, 14, 45, 54, 63, 71, 72]. Prior works have noticed that vanilla interpolating networks are sensitive to label noise (e.g. Figure 1 in Zhang et al. [71], and Belkin et al. [9]), and there are many works on making networks more robust to label noise via modifications to the training procedure or objective [45, 54, 63, 72]. In contrast, we claim this sensitivity to label noise is not necessarily a problem to be fixed, but rather a consequence of a stronger property: distributional generalization.

**Conditional Density Estimation.** Our density calibration property is similar to the guarantees of a conditional density estimator. More specifically, Conjecture 1 states that an interpolating classifier *samples* from a distribution approximating the conditional density of $p(y|x)$ in a certain sense. Conditional density estimation has been well-studied in classical nonparametric statistics (e.g. the Nadaraya–Watson kernel smoother [41, 66]). However, these classical methods behave poorly in high-dimensions, both in theory and in practice. There are some attempts to extend these classical methods to modern high-dimensional problems via augmenting estimators with neural networks (e.g. Rothfuss et al. [55]). Random forests have also been known to exhibit properties similar to conditional density estimators. This has been formalized in various ways, often only with asymptotic guarantees [5, 39, 52].

No prior work that we are aware of attempts to characterize the quality of the resulting density estimate via testable assumptions, as we do with our formulation of Conjecture 1. Finally, our motivation is not to design good conditional density estimators, but rather to study properties of interpolating classifiers — which we find happen to share properties of density estimators.

Feature Calibration (Conjecture 1) is also related to the concepts of calibration and multicalibration [26, 30, 48]. In our framework, calibration is implied by Feature Calibration for a specific set of partitions $L$ (determined by level sets of the classifier's confidence). However, we are not

concerned with a specific set of partitions (or "subgroups" in the algorithmic fairness literature) but we generally aim to characterize for which partitions Feature Calibration holds. Moreover, we consider only hard-classification decisions and not confidences, and we study only standard learning algorithms which are not given any distinguished set of subgroups/partitions in advance. Our notion of distributional generalization is also related to the notion of "distributional subgroup overfitting" introduced recently by Yaghini et al. [69] to study algorithmic fairness. This can be seen as studying distributional generalization for a specific family of tests (determined by distinguished subgroups in the population).

**Locality and Manifold Learning.** Our intuition for the behaviors in this work is that they arise due to some form of "locality" of the trained classifiers, in an appropriate space. This intuition is present in various forms in the literature, for example: the so-called called "manifold hypothesis," that natural data lie on a low-dimensional manifold (e.g. Narayanan and Mitter [44], Sharma and Kaplan [61]), as well as works on local stiffness of the loss landscape [19], and works showing that overparameterized neural networks can learn hidden low-dimensional structure in high-dimensional settings [6, 15, 21]. It is open to more formally understand connections between our work and the above.

**Note about Proper Scoring Rules:** If the loss function used in training is a *strictly-proper scoring rule* such as cross-entropy, then we may expect that in the limit of a large-capacity network and infinite data, training on samples $\{(x_i, y_i)\}$ will yield a good density estimate of $p(y|x)$ at the softmax layer. However, this is not what is happening in our experiments: First, our experiments consider the hard-decisions, not the softmax outputs. Second, we observe Conjecture 1 even in settings without proper scoring rules (kernel SVM and decision trees).

# B    Experimental Details

Here we describe general background, and experimental details common to all sections. Then we provide section-specific details below.

## B.1    Datasets

We consider the image datasets CIFAR-10 and CIFAR-100 [33], MNIST [34], Fashion-MNIST [68], CelebA [37], and ImageNet [56]. We normalize images to $x \in [0, 1]^{C \times W \times H}$.

We also consider tabular datasets from the UCI repository [16]. For UCI data, we consider the 121 classification tasks as standardized in Fernández-Delgado et al. [18]. Some of these tasks have very few examples, so we restrict to the 92 classification tasks from Fernández-Delgado et al. [18] which have at least 200 total examples.

## B.2    Models

We consider neural-networks, kernel methods, and decision trees.

### B.2.1    Decision Trees

We train interpolating decision trees using a growth rule from Random Forests [12, 31]: selecting a split based on a random $\sqrt{d}$ subset of $d$ features, splitting based on Gini impurity, and growing trees until all leafs have a single sample. This is as implemented by Scikit-learn [51] defaults with `RandomForestClassifier (n_estimators=1, bootstrap=False)`.

### B.2.2    Kernels

Throughout this work we consider classification via kernel regression and kernel SVM. For $M$-class classification via kernel regression, we follow the methodology in e.g. Belkin et al. [9], Rahimi and Recht [53], Shankar et al. [60]. We solve the following convex problem for training:

$$\alpha^* := \underset{\alpha \in \mathbb{R}^{N \times M}}{\operatorname{argmin}} ||K\alpha - y||_2^2 + \lambda \alpha^T K \alpha$$

where $K_{ij} = k(x_i, x_j)$ is the kernel matrix of the training points for a kernel function $k$, $y \in \mathbb{R}^{N \times M}$ is the one-hot encoding of the train labels, and $\lambda \geq 0$ is the regularization parameter. The solution

can be written

$$\alpha^* = (K + \lambda I)^{-1} y$$

670 which we solve numerically using SciPy `linalg.solve` [64]. We use the explicit form of all kernels
671 involved. That is, we do not use random-feature approximations [53], though we expect they would
672 behave similarly.

673 The kernel predictions on test points are then given by

$$g_\alpha(x) := \sum_{i \in [N]} \alpha_i k(x_i, x) \tag{11}$$

$$f_\alpha(x) := \operatorname*{argmax}_{j \in [M]} g_\alpha(x)_j \tag{12}$$

674 where $g(x) \in \mathbb{R}^M$ are the kernel regressor outputs, and $g(x) \in [M]$ is the thresholded classification
675 decision. This is equivalent to training $M$ separate binary regressors (one for each label), and taking
676 the argmax for classification. We usually consider *unregularized* regression ($\lambda = 0$), except in
677 Section 5.2.

678 For kernel SVM, we use the implementation provided by Scikit-learn [51] `sklearn.svm.SVC` with
679 a precomputed kernel, for inverse-regularization parameter $C \geq 0$ (larger $C$ corresponds to smaller
680 regularization).

681 **Types of Kernels.** We use the following kernel functions $k : \mathbb{R}^d \times \mathbb{R}^d \to \mathbb{R}_{\geq 0}$.

682 • Gaussian Kernel (RBF): $k(x_i, x_j) = \exp(-\frac{||x_i - x_j||_2^2}{2\tilde{\sigma}^2})$.

683 • Laplace Kernel: $k(x_i, x_j) = \exp(-\frac{||x_i - x_j||_2}{\tilde{\sigma}})$.

684 • Myrtle10 Kernel: This is the compositional kernel introduced by Shankar et al. [60]. We
685 use their exact kernel for CIFAR-10.

686 For the Gaussian and Laplace kernels, we parameterize bandwidth by $\sigma := \tilde{\sigma}/\sqrt{d}$. We use the
687 following bandwidths, found by cross-validation to maximize the unregularized test accuracy:

688 • MNIST: $\sigma = 0.15$ for RBF kernel.

689 • Fashion-MNIST: $\sigma = 0.1$ for RBF kernel. $\sigma = 1.0$ for Laplace kernel.

690 • CIFAR-10: Myrtle10 Kernel from Shankar et al. [60], and $\sigma = 0.1$ for RBF kernel.

### B.2.3 Neural Networks

692 We use 4 different neural networks in our experiments. We use a multi-layer perceptron, and three
693 different Residual networks.

694 **MLP:** We use a Multi-layer perceptron or a fully connected network with 3 hidden layers with 512
695 neurons in each layer. A hidden layer is followed by a BatchNormalization layer and ReLU activation
696 function.

697 **WideResNet:** We use the standard WideResNet-28-10 described in Zagoruyko and Komodakis [70].
698 Our code is based on this repository.

699 **ResNet50:** We use a standard ResNet-50 from the PyTorch library [50].

700 **ResNet18:** We use a modification of ResNet18 [29] adapted to CIFAR-10 image sizes. Our code is
701 based on this repository.

702 For Experiment 1 and Section 4, the hyperparameters used to train the above networks are given in
703 Table 1.

| | MLP | ResNet18 | WideResNet | ResNet50 |
|---|---|---|---|---|
| **Batchsize** | 128 | 128 | 128 | 32 |
| **Epochs** | 820 | 200 | 200 | 50 |
| **Optimizer** | Adam $(\beta_1 = 0.9, \beta_2 = 0.999)$ | SGD + Momentum (0.9) | SGD + Momentum (0.9) | SGD |
| **Learning rate (LR) schedule** | Constant LR = 0.001 | Inital LR= 0.05 scale by 0.1 at epochs $(80, 120)$ | Inital LR= 0.1 scale by 0.2 at epochs $(80, 120, 160)$ | Initial LR = 0.001, scale by 0.1 if training loss stagnant for 2000 gradient steps |
| **Data Augmentation** | Random flips + RandomCrop(32, padding=4) | | | |
| **CIFAR-10 Error** | $\sim 37\%$ | $\sim 8\%$ | $\sim 4\%$ | N/A |

Table 1: Hyperparameters used to train the neural networks and their errors on the unmodified CIFAR-10 dataset

## C Feature Calibration: Appendix

### C.1 A guide to reading the plots

All the experiments in support of Conjecture 1 involve various quantities which we enumaerate here

1. Inputs $x$: Each experiment involves inputs from a standard dataset like CIFAR-10 or MNIST. We use the standard train/test splits for every dataset.

2. Distinguishable feature $L(x)$: This feature depends only on input $x$. We consider various features like the original classes itself, a superset of classes (as in coarse partition) or some secondary attributes (like the binary attributes provided with CelebA)

3. Output labels $y$: The output label may be some modification of the original labels. For instance, by adding some type of label noise, or a constructed binary task as in Experiment 1

4. Classifier family $F$: We consider various types of classifiers like neural networks trained with gradient based methods, kernel and decision trees.

In each experiment, we are interested in two joint densities $(y, L(x))$, which depends on our dataset and task and is common across train and test, and $(f(x), L(x))$ which depends on the interpolating classifiers outputs on the *test* set. Since $y, L(x)$ and $f(x)$ are discrete, we will look at their discrete joint distributions. We sometimes refer to $(y, L(x))$ as the train joint density, as at interpolation $(y, L(x)) = (f(x), L(x))$ for all training inputs $x$. We also refer to $(f(x), L(x))$ as the test density, as we measure this only on the test set.

### C.2 Experiment 1

**Experimental details:** We now provide further details for Experiment 1. We first construct a dataset from CIFAR-10 that obeys the joint density $(y, L(x))$ shown in Figure 1 left panel. We then train a WideResNet-28-10 (WRN-28-10) on this modified dataset to zero training error. The network is trained with the hyperparameters described in Table 1. We then observe the joint density $(f(x), L(x))$ on the test images and find that the two joint densities are close as shown in Figure 5.

We now consider a modification of this experiment as follows:

**Experiment 2.** *Consider the following distribution over images $x$ and binary labels $y$. Sample $x$ as a uniformly random CIFAR-10 image, and sample the label as $p(y|x) = Bernoulli(\texttt{CIFAR\_Class(x)}/10)$. That is, if the CIFAR-10 class of $x$ is $k \in \{0, 1, \ldots 9\}$, then the label is $1$ with probability $(k/10)$ and $0$ otherwise. Figure 5 shows this joint distribution of $(x, y)$. As before, train a WideResNet to 0 training error on this distribution.*

In this experiment too, we observe that the train and test joint densities are close as shown in Figure 5.

Now, we repeat the same experiment, but with an MLP instead of WRN-28-10. The training procedure is described in Table 1. This MLP has an error on $37\%$ on the original CIFAR-10 dataset.

Since this MLP has poor accuracy on the original CIFAR-10 classification task, it does not form a distinguishable partition for it. As a result, the train and test joint densities (Figure 6) do not match as well as they did for WRN-28-10.

### C.3 Constant Partition

Conjecture 1 states that the marginal distribution of class labels for any interpolating classifier $f(x)$ is close to the true marginals $p(y)$. To show this, we construct a dataset based on CIFAR-10 that has class-imbalance. For class $k \in \{0...9\}$, sample $(k + 1) \times 500$ images from that class. This will give us a dataset where classes will have marginal distribution $p(y = \ell) \propto \ell + 1$ for classes $\ell \in [10]$, as shown in Figure 2. We do this both for the training set and the test set, to keep the distribution $\mathcal{D}$ fixed.

We then train a variety of classifiers (MLPs, RBF Kernel, ResNets) to interpolation on this dataset, which have varying levels of test errors (9-41%). The class balance of classifier outputs on the (rebalanced) test set

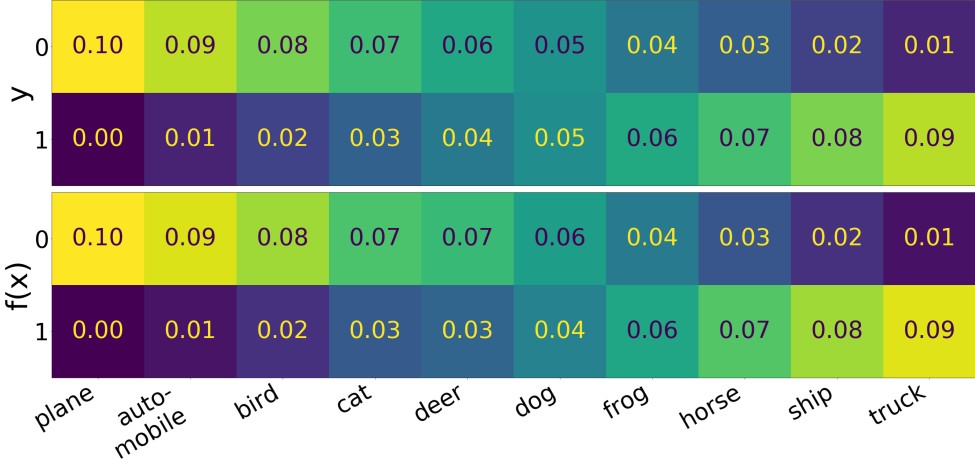

Figure 5: **Distributional Generalization in Experiment 2.** Joint densities of the distributions involved in Experiment 2. The top panel shows the joint density of labels on the train set: $(\texttt{CIFAR\_Class(x)}, y)$. The bottom panels shows the joint density of classifier predictions on the test set: $(\texttt{CIFAR\_Class(x)}, f(x))$. Distributional Generalization claims that these two joint densities are close.

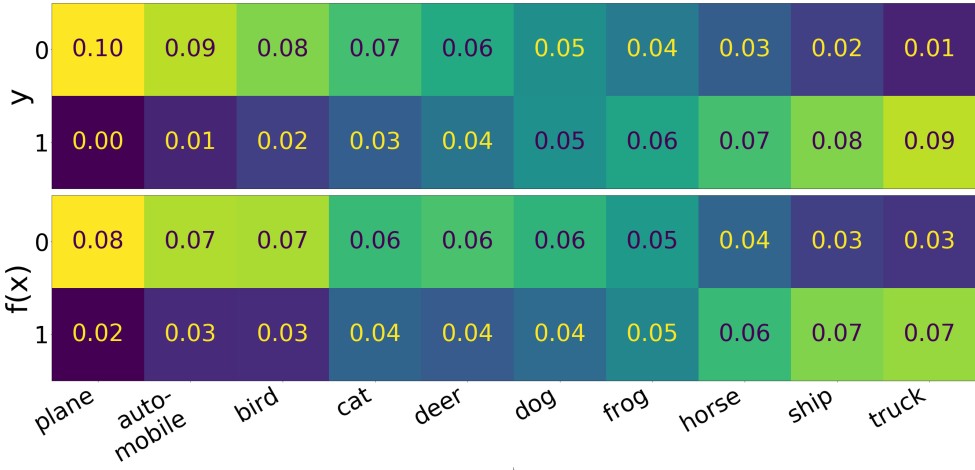

Figure 6: Joint density of $(y, \text{Class}(x))$, top, and $(f(x), \text{Class}(x))$, bottom, for test samples $(x, y)$ from Experiment 2 for an MLP.

## C.4    Class Partition

### C.4.1    Neural Networks and CIFAR-10

We now describe details for the experiments in Figures 2A and 3A. A WRN-28-10 achieves an error of $4\%$ on CIFAR-10. Hence, the original labels in CIFAR-10 form a distinguishable partition for this dataset. To demonstrate that Conjecture 1 holds, we consider different structured label noise on the CIFAR-10 dataset. To do so, we apply a variety of confusion matrices to the data. That is, for a confusion matrix $C : 10 \times 10$ matrix, the element $c_{ij}$ gives the joint density that a randomly sampled image had original label $j$, but is flipped to class $i$. For no noise, this would be an identity matrix.

We begin by a simple confusion matrix where we flip only one class $0 \rightarrow 1$ with varying probability $p$. Figure 7A shows one such confusion matrix for $p = 0.4$. We then train a WideResNet-28-10 to zero train error on this dataset. We use the hyperparameters described in B.2 We find that the classifier outputs on the test set closely track the confusion matrix that was applied to the distribution. Figure 7C shows that this is independent of the value of $p$ and continues to hold for $p = [0, 1]$.

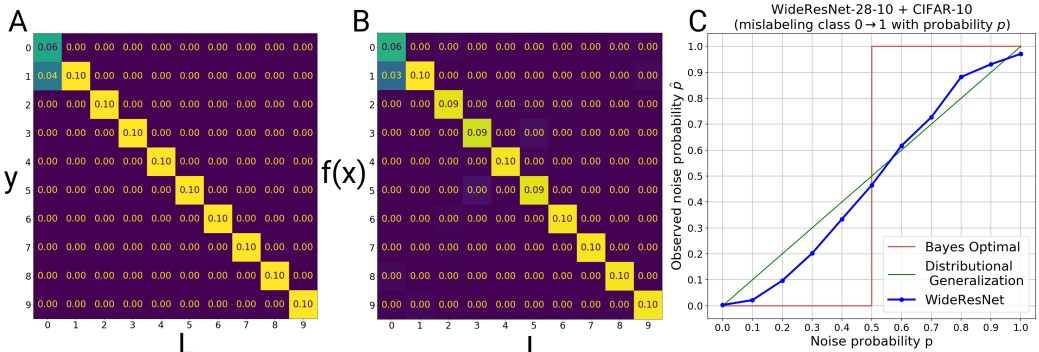

Figure 7: **Feature Calibration with original classes on CIFAR-10**: We train a WRN-28-10 on the CIFAR-10 dataset where we mislabel class $0 \rightarrow 1$ with probability $p$. (A): Joint density of the distinguishable features $L$ (the original CIFAR-10 class) and the classification task labels $y$ on the train set for noise probability $p = 0.4$. (B): Joint density of the original CIFAR-10 classes $L$ and the network outputs $f(x)$ on the test set. (C): Observed noise probability in the network outputs on the test set (the $(1, 0)$ entry of the matrix in B) for varying noise probabilities $p$

To show that this is not dependent on the particular class used, we also show that the same holds for a random confusion matrix. We generate a sparse confusion matrix as follows. We set the diagonal to $0.5$. Then, for every class $j$, we pick any two random classes for and set them to $0.2$ and $0.3$. We train a WRN-28-10 on it and report the test confusion matrix. The resulting train and test densities are shown in Figure 2A. As expected, the train and test confusion matrices are close, and share the same sparsity pattern.

### C.4.2   Decision Trees

Figure 8 shows a version of this experiment for decision trees on the molecular biology UCI task. The molecular biology task is a 3-way classification problem: to classify the type of a DNA splice junction (donor, acceptor, or neither), given the sequence of DNA (60 bases) surrounding the junction. We add varying amounts of label noise that flips class 2 to class 1 with a certain probability, and we observe that interpolating decision trees reproduce this same structured label noise on the test set.

Similar results hold for decision trees; here we show experiments on two UCI tasks: `wine` and `mushroom`.

The `wine` task is a 3-way classification problem: to identify the cultivar of a given wine (out of 3 cultivars), given 13 physical attributes describing the wine. Figure 9 shows an analogous experiment with label noise taking class 1 to class 2.

The `mushroom` task is a 2-way classification problem: to classify the type of edibility of a mushroom (edible vs poisonous) given 22 physical attributes (e.g. stalk color, odor, etc). Figure 10 shows an analogous experiment with label noise flipping class 0 to class 1.

### C.5   Multiple Features

Conjecture 1 states that the network should be automatically calibrated for all distinguishable features, without any explicit labels for them. To verify this, we use the CelebA dataset [37], containing images with various labelled binary attributes per-image ("male", "blond hair", etc). Some of these attributes form a distinguishable feature for ResNet50 as they are learnable to high accuracy [32]. We pick one of hard attributes as the target classification task. We train a ResNet-50 to predict the attribute {Attractive, Not Attractive}. We choose this attribute because a ResNet-50 performs poorly on this task (test error $\sim 20\%$) and has good class balance. We choose an attribute with poor generalization because the conjecture would hold trivially for if the network generalizes well. We initialize the network with a pretrained ResNet-50 from the PyTorch library [50] and use the hyperparameters described in Section B.2 to train on this attribute. We then check the train/test joint density with various other attributes like Male, Wearing Lipstick etc. Note that the network is not given any label

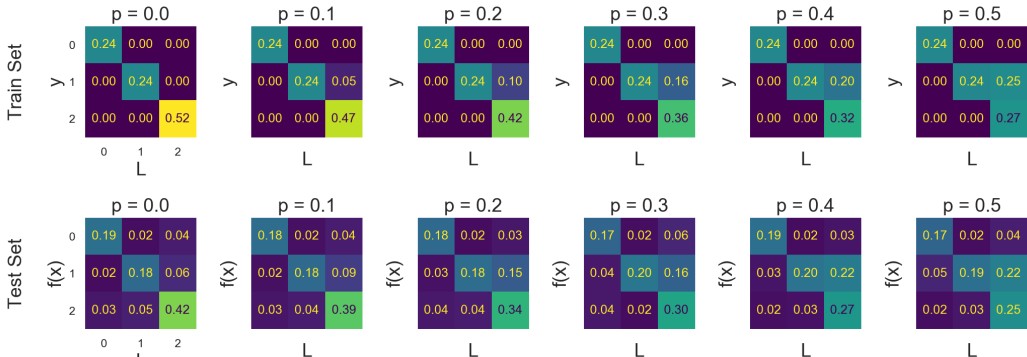

Figure 8: **Feature Calibration for Decision trees on UCI (molecular biology).** We add label noise that takes class 2 to class 1 with probability $p \in [0, 0.5]$. The top row shows the confusion matrix of the true class $L(x)$ vs. the label $y$ on the train set, for varying levels of noise $p$. The bottom row shows the corresponding confusion matrices of the classifier predictions $f(x)$ on the test set, which closely matches the train set, as predicted by Conjecture 1.

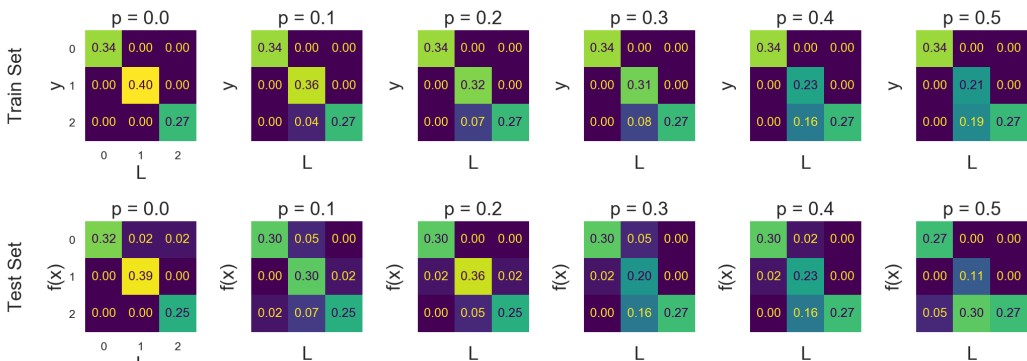

Figure 9: Decision trees on UCI (wine). We add label noise that takes class 1 to class 2 with probability $p \in [0, 0.5]$. Each column shows the test and train confusion matrices for a given $p$. Note that this decision trees achieve high accuracy on this task with no label noise (leftmost column). We plot the empirical joint density of the train set, and not the population joint density of the train distribution, and thus the top row exhibits some statistical error due to small-sample effects.

information for these additional attributes, but is calibrated with respect to them. That is, the network says $\sim 30\%$ of images that have 'heavy makeup' will be classified as 'Attractive', even if the network makes mistakes on which particular inputs it chooses to do so. In this setting, the label distribution is deterministic, and not directly dependent on the distinguishable features, unlike the experiments considered before. Yet, as we see in Figure 11, the classifier outputs are correctly calibrated for each attribute. Loosely, this can be viewed as the network performing 1NN classification in a metric space that is well separated for each of these distinguishable features.

## C.6 Coarse Partition

We now consider cases where the original classes do not form a distinguishable partition for the classifier in consideration. That is, the classifier is not powerful enough to obtain low error on the original dataset, but can perform well on a coarser division of the classes.

To verify this, we consider a division of the CIFAR-10 classes into Objects {airplane, automobile, ship, truck} vs Animals {cat, deer, dog, frog}. An MLP trained on this problem has low error ($\sim 8\%$), but the same network performs poorly on the full dataset ($\sim 37\%$ error). Hence, Object vs Animals forms a distinguishable partition with MLPs. In Figure 12a, we show the results of training an MLP

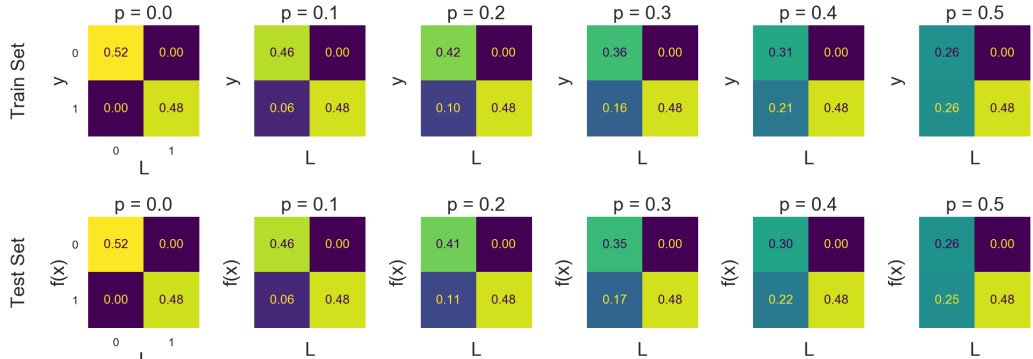

Figure 10: Decision trees on UCI (mushroom). We add label noise that takes class 0 to class 1 with probability $p \in [0, 0.5]$. Each column shows the test and train confusion matrices for a given $p$. Note that this decision trees achieve high accuracy on this task with no label noise (leftmost column).

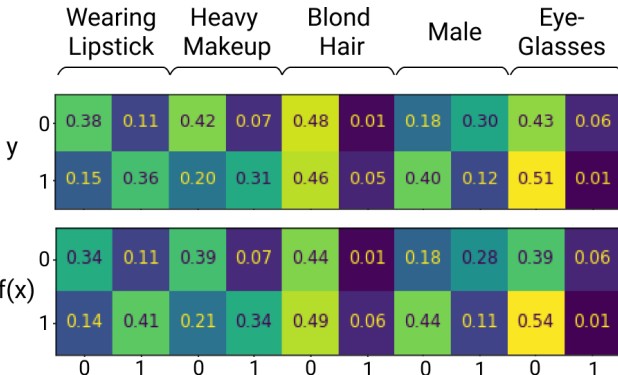

Figure 11: **Feature Calibration for multiple features on CelebA**: We train a ResNet-50 to perform binary classification task on the CelebA dataset. The top row shows the joint distribution of this task label with various other attributes in the dataset. The bottom row shows the same joint distribution for the ResNet-50 outputs on the test set. Note that the network was not given any explicit inputs about these attributes during training.

on the original CIFAR-10 classes. We see that the network mostly classifies objects as objects and animals as animals, even when it might mislabel a dog for a cat.

We perform a similar experiment for the RBF kernel on Fashion-MNIST, with partition {clothing, shoe, bag}, in Figure 12b.

**ImageNet experiment.** In Table 2 we provide results of the terrier experiment in the body, for various ImageNet classifiers. We use publicly available pretrained ImageNet models from this repository, and use their evaluations on the ImageNet test set.

## C.7 Discussion: Proper Scoring Rules

Here we distinguish the density-estimation of Conjecture 1 from another setting where density estimation occurs. If $\ell(\widehat{p}, y)$ is a *strictly-proper scoring rule*[3] on predicted distribution $\widehat{p} \in \Delta(\mathcal{Y})$ and sample $y \in \mathcal{Y}$, then the population minimizer of $\ell(F(x), y)$ is exactly the conditional density $F(x) = p(y|x)$. That is,

$$p(y|x) = \underset{F: \mathcal{X} \to \Delta(\mathcal{Y})}{\operatorname{argmin}} \ \underset{(x,y) \sim p}{\mathbb{E}} \left[ \ell(F(x), y) \right]$$

---

[3]See [23] for a survey of proper scoring rules.

| Model | AlexNet | ResNet18 | ResNet50 | BagNet8 | BagNet32 |
|---|---|---|---|---|---|
| ImageNet Accuracy | 0.565 | 0.698 | 0.761 | 0.464 | 0.667 |
| Accuracy on dogs | 0.588 | 0.729 | 0.793 | 0.462 | 0.701 |
| Accuracy on terriers | 0.572 | 0.704 | 0.775 | 0.421 | 0.659 |
| Accuracy for binary {dog/not-dog} | 0.984 | 0.993 | 0.996 | 0.972 | 0.992 |
| Accuracy on {terrier/not-terrier} among dogs | 0.913 | 0.955 | 0.969 | 0.876 | 0.944 |
| Fraction of real-terriers among dogs | 0.224 | 0.224 | 0.224 | 0.224 | 0.224 |
| **Fraction of predicted-terriers among dogs** | 0.209 | 0.222 | 0.229 | 0.192 | 0.215 |

Table 2: ImageNet classifiers are calibrated with respect to dogs: All classifiers predict terrier for roughly $\sim 22\%$ of all dogs (last row), though they may mistake which specific dogs are terriers.

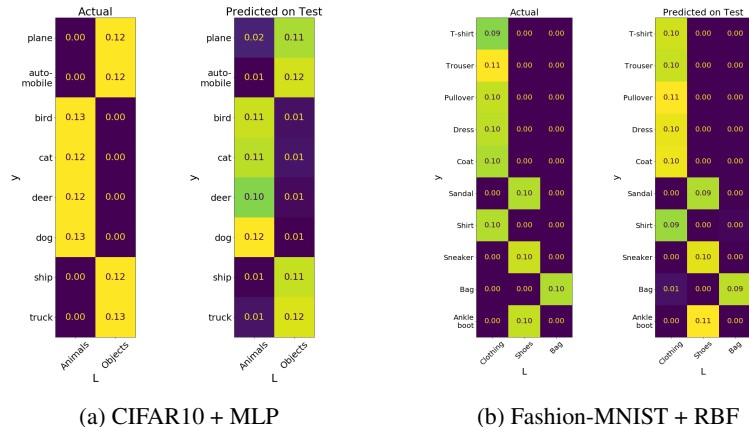

(a) CIFAR10 + MLP    (b) Fashion-MNIST + RBF

Figure 12: Coarse partitions as distinguishable features: We consider a setting where the original classes are not distinguishable, but a superset of the classes are distinguishable.

This suggests that in the limit of large-capacity network and very large data (to approximate population quantities), training neural nets with cross-entropy loss on samples $(x, y)$ will yield a good density estimate of $p(y|x)$ at the softmax layer. However, this is not what is happening in our experiments. First, our experiments consider the hard-thresholded classifier, i.e. the argmax of the softmax layer. If the softmax layer itself was close to $p(y|x)$, then the classifier itself will be close to $\mathrm{argmax}_y\, p(y|x)$ – that is, close to the optimal classifier. However, this is not the case (since the classifiers make significant errors). Second, we observe Conjecture 1 even in settings where we train with non-proper scoring rules (e.g. kernel regression, where the classifier does not output a probability).

# D   Nearest-Neighbor Proofs

## D.1   Feature Calibration Property

*Proof of Theorem 1.* Recall that $L$ being an $(\varepsilon, \mathrm{NN}, \mathcal{D}, n)$-distinguishable partition means that nearest-neighbors works to classify $L(x)$ from $x$:

$$\Pr_{\substack{\{x_i, y_i\} \sim \mathcal{D}^n \\ S = \{(x_i, L(x_i)\} \\ x, y \sim \mathcal{D}}} [\mathrm{NN}_S^{(y)}(x) = L(x)] \geq 1 - \varepsilon \tag{13}$$

Now, we have

$$\{(\mathrm{NN}_S^{(y)}(x), L(x))\}_{\substack{S \sim \mathcal{D}^n \\ x, y \sim \mathcal{D}}} \equiv \{(\widehat{y_i}, L(x))\}_{\substack{S \sim \mathcal{D}^n \\ \widehat{x_i}, \widehat{y_i} \leftarrow \mathrm{NN}_S(x) \\ x, y \sim \mathcal{D}}} \tag{14}$$

$$\approx_\varepsilon \{(\widehat{y_i}, L(\widehat{x_i}))\}_{\substack{S \sim \mathcal{D}^n \\ \widehat{x_i}, \widehat{y_i} \leftarrow \mathrm{NN}_S(x) \\ x, y \sim \mathcal{D}}} \tag{15}$$

$$\approx_\delta \{(\widehat{y_i}, L(\widehat{x_i}))\}_{\widehat{x_i}, \widehat{y_i} \sim \mathcal{D}} \tag{16}$$

Line (15) is by distinguishability, since $\Pr[L(x) \neq L(\widehat{x_i})] \leq \varepsilon$. And Line (16) is by the regularity condition. $\qquad \square$

# E  Non-interpolating Classifiers: Appendix

Here we give an additional example of distributional generalization: in kernel SVM (as opposed to kernel regression, in the main text).

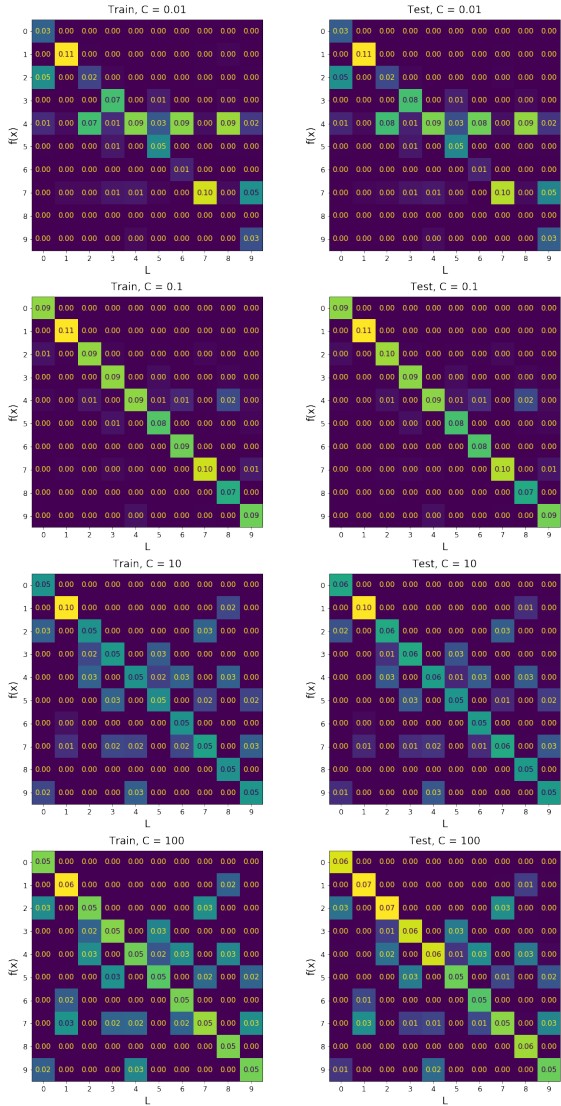

Figure 13: **Distributional Generalization.** Train (left) and test (right) confusion matrices for kernel SVM on MNIST with random sparse label noise. Each row corrosponds to one value of inverse-regularization parameter $C$. All rows are trained on the same (noisy) train set.

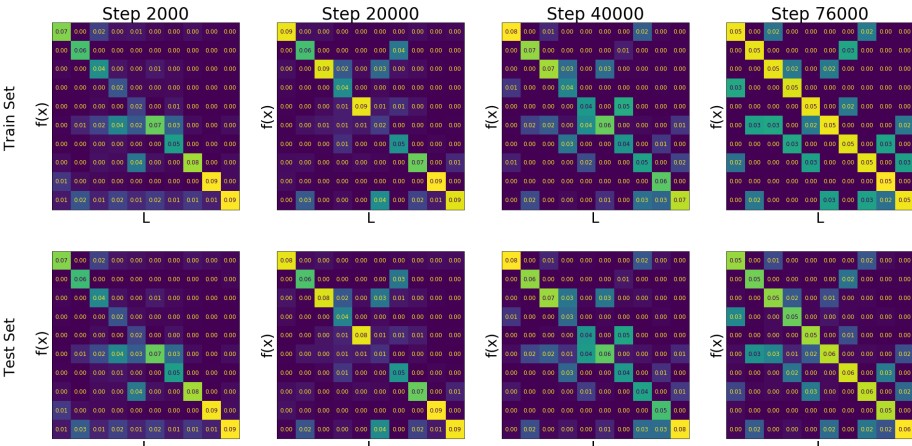

Figure 14: **Distributional Generalization for WideResNet on CIFAR-10.** We apply label noise from a random sparse confusion to the CIFAR-10 train set. We then train a single WideResNet28-10, and measure its predictions on the train and test sets over increasing train time (SGD steps). The top row shows the confusion matrix of predictions $f(x)$ vs true labels $L(x)$ on the train set, and the bottom row shows the corresponding confusion matrix on the test set. As the network is trained for longer, it fits more of the noise on the train set, and this behavior is mirrored almost identically on the test set.