# OpenReview forum: "Distributional Generalization: Characterizing Classifiers Beyond Test Error"
_NeurIPS.cc/2021/Conference — NeurIPS 2021 Submitted_

### Official Review · Reviewer_fCRG · 2021-07-14

**Rating:** 6
**Confidence:** 4

**Summary:**

The paper empirically characterizes a fine-grained generalization property of interpolating learning algorithms, and give a conjecture that formalizes the observed phenomenon. Roughly, the paper observes that the generalization properties of an interpolating classifier can be quite distinct for different sample subpopulations with the same label, whenever the label noise imposed on these subpopulations during the training phase is distinct. Authors conjecture that the generalization properties may be uniform only for samples belonging to a smaller "distinguishable" clusters of the input data space.

**Ethical Concerns:**

I cannot think of any relevant ethical issue.

**Limitations And Societal Impact:**

The limitation is clearly described, in my opinion. This work mainly discusses what is implicitly taking place in the standard learning practices, instead of giving a new one. I do not think there is any "potential negative impact" that should be discussed.

**Main Review:**

Post-rebuttal: I thank authors for their detailed response. Most of the authors' response made a perfect sense. One thing that prevented me from raising the score further is that I am still not sure about is whether the conjecture itself is meaningful or useful. In fact, after reading the response to my first point (Converse...), I am slightly more pessimistic about the possibility that "distinguishability" and "feature calibration" are tightly connected concepts (say, in a causal sense). I am curious if there are alternate (and perhaps more useful) conjectures that can help explain the phenomenon. In other words, I think the paper should benefit from having an extensive discussion that justifies why the proposed conjecture based on distinguishability is nontrivial, useful, and better than other possible alternatives.


---

__What I like.__ The reported empirical observation is intriguing to some degree, and (in my opinion) is worthy of documenting and formalizing. Authors validate the observation under a variety of scenarios in Section 1 & 4 (and also appendices), which helped convincing me of the existence of such phenomenon. As the authors noted, establishing a theoretical framework that can explain the depicted phenomenon may be an interesting open problem (perhaps we can phrase problems in a minimax style with multiple tasks?). The clarity of the writing is well above average, while there are several points that can be clarified further. I also liked the theoretical contributions, which could help readers grasp the idea of how such conjectures can be proved.

__Checking/refining the conjecture.__ On the one hand, I find that the provided empirical observations support the conjecture well. On the other hand, I think that one can even further strengthen the argument by trying to provide an evidence supporting that the conditions described in the conjecture is indeed essential for the distributional similarity. I am actually very curious whether the "distinguishable features" are really essential. Is there a case where the distributional similarity does not hold for indistinguishable features, which is perhaps more informative than the perfectly random case? Although this is not really necessary if one is interested only in the validity of the conjecture, having such evidence will help us understand how non-vacuous/"tight" the conjecture is.

__Clarifying the terminology: Interpolating.__ I think the term "interpolating classifier" can be clarified further in the text. Although the terminology is now (sort of) popularly used in the literature, I think still they need to be properly defined at least once in the main text, as the modern use of this terminology are often wider than what is used to mean in the classic kernel-based literature.

__Clarifying the terminology: Distributional Closeness.__ I am curious in what sense the notion of "distributional closeness" introduced in Section 2 is different from the widely used notion of integral probability metrics (IPMs). If they are identical, sticking to the existing notation would help reduce the confusion.

__Practical implication.__ While I mostly liked the theoretical implications illustrated in Section 1.3, I wonder if authors could give an idea why capturing such fine-grained generalization properties may be important. For instance, could there be any implications to the fairness or shortcut learning?

**Time Spent Reviewing:**

24

---

> ### Author Response · Authors · 2021-08-09
> **Response to Reviewer fCRG**
>
> Thank you for your detailed review and thoughtful questions. We appreciate your comments that our work is "intriguing" and "worthy of documenting and formalizing."
> We respond to your questions below, and are happy to continue the discussion if any points are unclear. If your concerns are addressed, and the work is strong enough to be published, we kindly ask that you consider increasing your score to "accept."
>
> **Re. Converse to the conjecture:** This is a very interesting question. One reason we did not state a converse to our conjecture is that technically, the formal converse is false: it is not true that "indistinguishable" features always induce distributions which are "far" (in the feature-calibration sense).
> The issue is subtle: there are cases where "indistinguishable" features lead to similar joint distributions, and cases where they lead to dissimilar joint distributions. Thus, "indistinguishability" is not a sufficiently nice condition to pin down behaviors, unlike "distinguishability". Here is an explicit example of these 2 cases:
>
> Case 1: The feature L(x) is a random function, independent of the label Y.
> In this case, L(x) is indistinguishable (since random functions are hard to learn), but Feature Calibration still holds w.r.t L(x). This occurs because, since L is random and independent of both Y and f, feature calibration holds almost trivially, with small \epsilon (it just implies that the marginal distribution of f(x) is correct). This example also works if L(x) is not a random function, but some other "hard-to-learn" function that's statistically independent of labels Y.
>
> Case 2: The feature L(x) is a "hard-to-learn" function that's correlated with the label Y.
> For example, suppose L(x) = Y(x) = hash(x), where `hash` is some hard-to-learn function.
> Note that the feature L is exactly equal to the label Y in this example.
> In this case, L(x) is indistinguishable, and Feature Calibration fails to hold (i.e. does not hold with small epsilon).
>
> These two different conclusions highlights the fact that "indistinguishability" alone is not enough to determine whether distributions are close and far. Rather, we must somehow account for the interaction of the *labels* Y with the feature L(x). Note that this was not necessary in our conjecture: distinguishability of L(x) does not require any assumptions on the labels Y.
>
> However, although the formal converse to our conjecture is false, we agree that exploring conditions under which the distributions are "far" is a very interesting question.
>
> As a final point, note that the \epsilon upper-bound on distributional closeness in our Feature Calibration conjecture is in fact tight, and cannot be improved. That is, for all epsilon, there exist \epsilon-distinguishable features for which the distributions are exactly \epsilon-far in TV distance. The example which saturates this bound is when L(x) = Y(x), and the labels Y can only be learnt to \epsilon test error. This is similar to Case 2 above.
>
>
> **Re. terminology:** Thank you for the comments; we will clarify the terminology of "interpolating" and integral probability metrics (IPMs).
> As you note, our notion of distributional closeness is exactly identical to IPMs (and we had a reference to this, but had to remove it due to space constraints).
>
> Regarding why we did not choose to use the IPM notation: This same concept of distributional closeness has been studied in various communities under different names. We chose to use the notation of "distinguishers" from the theoretical cryptography literature, because it is more aligned with our intuitions about this particular problem (e.g. we have the intuition of various computational "tests" which we apply to distributions). To be clear, our notation of epsilon-closeness is inherited from the cryptography literature, and not a notation we invented.
>
> In any case, in the revision we will certainly be more explicit about the fact that this concept is equivalent to IPMs, and that the notation is inherited from complexity theory.
> (We also appreciate the suggestion to use notation that may be more familiar to NeurIPS readers).
>
> **Re. Practical implications**: Thank you for this question, which led us to think more about certain fairness implications.
>
> First, one application we consider important (though not about fairness) is described in the "Coarse partition" paragraph on L252. The idea is, Feature Calibration can tell us that even classifiers which "look bad" with respect to test error can have certain "good behaviors" which are relevant in applications.
> For example, AlexNet has only 56% test accuracy on the 1000-class problem of ImageNet, which includes 100+ breeds of dogs.
> This seems like poor accuracy. However, suppose we ask: if we input an image of a dog, will it at least output the label of *some* dog breed? (As opposed to a non-dog label, e.g. houseplant).
> The Feature Calibration conjecture implies that this is indeed the case, because dog/not-dog is a distinguishable feature.
>
> Thus, our conjecture tells us more about *what kinds of errors to expect* from models. In this case, we know that even though we have 44% error overall, these errors occur mostly *within* the dog or not-dog subgroups, and not *across* these groups. Understanding when models make errors is an important topic, which has received much attention in practice.
>
>
>
> One potential fairness application is below.
>
> **Fairness Application**:
> Suppose we are in a classification setting, where we wish to classify labels Y given input X. Moreover, there is a sensitive attribute R(x) \in {0, 1}, such as Race.
> Suppose that (Y, R) are statistically independent in the population distribution. That is, the true labels Y are independent of race R.
> Now we ask, will our trained classifier have outputs F(x) that are also independent of race R?
> This is far from trivial -- it may or may not hold, depending on the classifier. Moreover, it may be a desirable property from the fairness perspective.
>
> Our Feature Calibration conjecture implies that if our interpolating classifier is powerful enough to "distinguish" race (in the "distinguishable feature" sense), then it will satisfy the above statistical independence. That is, the outputs F(x) will be statistically independent of R(x) at test time.
> This suggests that there is benefit to using a classifier which is capable of distinguishing Race, even if the ground-truth labels are independent of race.
>
> It is also possible to construct toy settings where the converse is true: where "weak" classifiers produce outputs F(x) which are dependent on Race, even when the ground-truth is not. (Intuitively, these are in cases where the classifier depends on some "easy" feature, which is correlated to both race and the output labels).
>
> In summary, our conjectures imply that "powerful classifiers preserve statistical independence of sensitive attributes", in the formal sense described above.

---

> > ### Author Response · Authors · 2021-09-18
> > **Response to post-rebuttal reviewer edits**
> >
> > Regarding the reviewer's post-rebuttal response:
> >
> > We first emphasize that our Feature Calibration conjecture is in fact tight, with matching upper and lower bounds, as we described in our response. A conjecture with tight bounds is already an extremely strong statement; the lack of a converse should not be seen as a weakness. Indeed, in science and mathematics, there are many interesting theorems that do not have converses. This is not a weakness.
> >
> > We do not understand the comment about "connected concepts in a causal sense." The language of causality is not appropriate to discuss the behaviors we consider. We also remark that many papers claiming to present a "causal understanding of generalization" misuse the concept of causality, and should not be taken as exemplar. We are happy to elaborate on the irrelevance of causality to our setting if the reviewer or the AC desires.
> >
> > Finally, we strongly disagree with evaluating the merits of scientific work based on its practical "usefulness." Scientific inquiry and industrial application have very different motivations and goals. Moreover, in the history of science, the true "usefulness" of a scientific discovery is rarely realized at the time of discovery.
> >
> > However, if the reviewer is keen on applications, note that there is already a follow-up paper to our work, which shows how to construct a practical out-of-distribution uncertainty estimator based on our conjectures. This was done by other authors, but directly acknowledges building on our paper on arxiv. (We cannot cite it here due to anonymity).
> >
> > To summarize: the reviewer asks why our conjecture is "nontrivial, useful, and better than other possible alternatives". We answer:
> > 1. It is **nontrivial** because it is a new empirical behavior that had not been noticed and formalized before. It is not implied by any existing conjectures or theorems in theory or in practice. Moreover, it is in direct conflict with some prevailing literature on interpolating classifiers (eg "benign overfitting") -- and thus it will be a significant update to the community.
> > 2. Scientific work should not be judged on practical **usefulness.**
> > 3. It is "better than other possible alternatives" because the quantitative form of the conjecture is tight (matching upper and lower bounds). More importantly, scientific progress happens in steps. There may indeed be better and more precise conjectures out there, as there is always more to be understood. We hope that this paper leads to many more papers -- it is only the beginning of this area, and we cannot expect to understand all the questions.
> >
> > We have presented a conjecture that is nontrivial, surprising, empirically true, has deep implications in the areas of generalization and interpolation, and has already led to follow-up practical work. This should be more than sufficient to be accepted to NeurIPS. We are surprised that the reviewer disagrees.

---

### Official Review · Reviewer_fE8g · 2021-07-16

**Rating:** 4
**Confidence:** 4

**Summary:**

The authors consider interpolating classifiers, i.e. classifiers that interpolate the training data by selecting the correct class for training samples, often performing poorly in terms of generalization error. The authors propose a new notion of generalization, i.e. the Distributional Generalization. Such a notion does not focus on the average error over the inputs, and subsumes both classical generalization and the conjecture that the outputs of classifiers match the statistics of their training distribution when conditioned on certain subgroups. The authors formally introduce and experimentally validate a conjecture to specify such subgroups. While the conjecture is proven for 1-nearest-neighbor classifiers, empirical evidence is provided for other classifiers; such as kernel machines and neural networks.

**Ethical Concerns:**

There is no ethical concerns

**Limitations And Societal Impact:**

The authors take into the scientific limitations of their work in section 3.4. Further improvements can be implemented by answering the questions and doubts raised in the Main Review.

**Main Review:**

== Main comments ==

- The paper is overall well written and the topics are clearly exposed. Illustrative examples are effectively presented to further clarify the problem at hand.
- The presented work tackles a timely and relevant problem in machine learning, drawing ideas from many related areas of research, such as output distribution calibration, overparametrization, and benign overfitting. The presented results are worthy but not completely surprising.
- The paper addresses a very much similar problem than :
Vapnik, V. & Izmailov, R.. (2020). Complete statistical theory of learning: learning using statistical invariants. Proceedings of the Ninth Symposium on Conformal and Probabilistic Prediction and Applications, in PMLR 128:4-40
However, this problem is known to below to the class of  ill-posed problems. How does the current work related to the above mentioned reference? How does it overcome the ill-posed problem.
- In eq. (1) the notation is a bit confusing. Are TrainSet and TestSet the distributions from which the samples of these sets are drawn (but then D_tr, D_te are used to mean the same), or rather the sets themselves (and then ~ should be replaced with ∈)?
- One of the two main contributions is thoroughly expressed in section 3.2. This is indeed a very interesting observation. However it looks closely related to the problem defined in section 2 of [26].
Could the authors further underline the value and originality of their contribution by discussing the connection between the two concepts?
- Although the feature calibration conjecture is fully proved to 1-nearest-neighbors classifiers, it is also true that it represents a very limited model. It seems that for more complex (and more commonly used) models, only empirical evidence can be provided so far, with no much theoretical formalization. Although this is not necessarily a negative aspect for works in progress and contributions which bring new ideas, the expectation set up by the writers in the introductory section are diminished by lack of formalization.
- The training procedure is sometimes indicated as Train_A, and sometimes as A. We suggest to choose one symbol and consistently use it throughout the whole paper.
- Have the authors thought of any potential issue related to identifying potentially sensitive subgroups L such as, ethnicity, high/low income, etc. instead of dogs, cats, etc. when using different
datasets?
- We would like to draw attention on the fact that, in line 101, the reference to the work in [7] does not seem to properly fit it. Indeed, in [7] the authors already explain that ‘benign overfitting’ is only characterized in specific cases and does not generally apply, making the claim in lines 103-104 appear a bit excessive. Reference [26] should be reported as the version officially published at ICML 2017.

**Time Spent Reviewing:**

5 hours

---

> ### Author Response · Authors · 2021-08-09
> **Response to Reviewer fE8g**
>
> Thank you for your valuable feedback and comments. We appreciate that you find the work "timely and relevant", and the paper "well written and the topics are clearly exposed."
> We have responded to your questions and concerns in detail below. We kindly ask that you consider increasing your score to "accept" if your concerns are addressed.
> We are happy to follow-up in further discussion if any concerns remain.
>
>
> **Regarding the work of Vapnik & Izmailov**:
> Our work is indeed somewhat related to the work of Vapnik & Izmailov, but there are important differences.
> First, and most importantly, we do not propose a new learning method. Instead, we study properties of existing methods, which had not been noticed before. Specifically, we consider standard training techniques of neural-networks, SVMs, kernels, and decision trees.
> In contrast, the work of Vapnik & Izmailov proposes new learning methods ("LUSI algorithms") in order to address certain problems.
>
> Both our works involve a related idea of using "predicates" to discuss distributional closeness (weak convergence in the Vapnik & Izmailov paper, and "feature calibration" in our paper). However, our work is about identifying which predicates hold "automatically" for a given learning method. In contrast, Vapnik & Izmailov's work is about how to construct a learning method which satisfies a given set of predicates, specified in advance. These are complementary goals: our work shows that classifiers can satisfy many more predicates than we explicitly ask for at training-time (for example, even if we only minimize the predicate of Training Error, our classifiers are "calibrated" with respect to many other predicates as well).
>
> Regarding the ill-posedness: This issue does not directly apply in our setting, for several reasons. The main one is that we do not ask for uniqueness (nor continuity). In our high-dimensional settings, we do not expect that the data will uniquely determine the conditional density. However, we are satisfied if we find an estimator which approximates the true density well enough *on-distribution* (i.e. with respect to some set of functionals). Moreover, our conjectures apply even for estimators which are not continuous, like decision trees.
>
>
> **Regarding comparison to Guo et al. [26]**:
> We discussed the relation to the "calibration" literature (including [26]) in lines 628-638 in the Appendix, and we elaborate further here.
> Our conjecture of "Feature Calibration" is significantly different from the notion of calibration in prior works (as [26]). Specifically, calibration in [26] means producing "confidence estimates" for each test point, and requiring that these estimates accurately reflect the true accuracies (when averaged over each confidence level-set).
>
> In our setting, our motivation is entirely different: we are not motivated by uncertainty estimation.
> We do not require classifiers to produce "confidences" of any kind -- classifiers only output hard classification decisions.
> Moreover, we consider partitioning the domain by many more subgroups than just level-sets of some function.
> Our conjecture then states that classifiers have the "right" behavior when conditioned on any such subgroup ("distinguishable feature").
>
> In summary, our Feature Calibration Conjecture is considerably more general than the standard notion of "calibration," and applies even in settings where classifiers do not have "confidences" defined. The standard notion of "calibration" can be seen as a special case of our conjecture, when applied to confidence level-sets as the "distinguishable feature" L(x).
>
> **Regarding "lack of formalization"**:
> We consider it important to distinguish between lack of formalism and lack of proofs.
> We have stated conjectures which are as mathematically formal as possible, modulo certain deep theoretical open problems (such as defining "natural distributions").
> We do not have proofs of these conjectures. However, we view the statement of these conjectures alone as a significant advance.
> There are many examples in the history of science where formally characterizing a behavior is the first step towards a deeper theoretical understanding. For example, in physics many "laws" were first stated as empirical facts, and only later derived from underlying principles (eg: Kepler's Laws, Hooke's Law, Ideal Gas Law, etc). This also occurs in machine learning, where many empirical phenomena were first discovered in practice, then later explained by theory. We view our work as another instance of this established method of scientific inquiry.
>
> We do hope that our work will eventually lead to a theoretical understanding of Distributional Generalization for neural networks and other models. But this is beyond the scope of the current paper. Indeed, we are very far from a theoretical understanding of even classical generalization, and so hoping to fully understand our new kind of generalization is ambitious.
>
> **Regarding the reference to Benign Overfitting [7]**:
> We agree that formally, none of our results contradict those of [7]. However, the conceptual message of our work is significantly different from [7].
>
> For example, the first sentence of the abstract of [7] includes the claim that "deep neural networks seem to predict well, even with a perfect fit to noisy training data."
> In contrast, the main message of our work is that fitting noisy training data actually *hurts* predictive performance in many realistic settings (since noise in the train set shows up as noise at test time).
> However, we are happy to adjust this wording; it is not our intention to be antagonistic.
>
> **Regarding potentially sensitive subgroups**:
> Our work may have applications in fairness, though it is not our primary focus, since we consider the behavior of classifiers on subgroups. For example, if some sensitive subgroup has lower-quality labels in the train set (e.g. higher label noise), then our conjectures imply that an interpolating classifier will also have higher error on this subgroup at test time (provided the sensitive subgroup is "distinguishable").
> We also describe another explicit fairness application in our response to reviewer fCRG.
>
> We stress that our work does not introduce any new methods; it only studies existing methods.
> Thus, while our work may help understand fairness issues in existing methods, we do not expect it will create any new fairness concerns which did not already exist.
>
> **Regarding notation**: Thank you for pointing out these notational issues, we will address them. For equation (1) specifically, we realize this equation is informal and thus may be unclear (as we pointed out in the footnote on pg2). We included Eq (1) in the Intro to give an intuitive sense of our conjectures, before we formally define them in Section 3. However, we will re-evaluate whether this trade-off of intuition for precision is confusing to readers.

---

> > ### Comment · Reviewer_fE8g · 2021-08-23
> > **Reply to author comments**
> >
> > I would like to thank the authors for the careful reading and response to my concerns. Some of these replies are convincing enough but others remind rather elusive. In particular, regarding the lack of theory and further understanding of the underlying statistical problem, including important connections with  Vapnik & Izmailov's work.
> >
> > At a high level, the statistical problem under consideration  is rather novel and may be well stated as follows: the problem consists in learning a partition function L (e.g., {dog, cat, horse. . . }) that maps the feature space X (in general continuous)  into M bins and satisfies that the distributions of (L(x),f(x)) and (L(x),y) are epsilon-close in TV-distance. In this sense, several interesting questions of theoretical nature with practical implications can be raised here:
> > - Is the partition function L  learnable from samples?
> > - How large M can be?
> > - Can M be increased with the size of the training set n?
> > - Is there any connexion with the complexity of the class of models of the classifier?
> > Unfortunately, very little information (mostly heuristic) is provided. More important, the problem at hand is very much related to the considered framework by Vapnik & Izmailov's work and I believe the authors did not manage to conveniently show why learning the distribution P(f(X) | L(X)) is a well stated problem compared to the more general case studied by Vapnik & Izmailov with respect to the distribution P(Y|X).  Obviously, the reason for this is because L(X) is discrete by definition and thus, it is possible (in principle) to learn P(f(X)|L(X)). However, this requires a formal proof to show the learnability of the function L() from dedicated samples, which is particular important for the considered framework in this paper. In addition, there are imprecise mathematical statements with non-conclusive  numerical validation, e.g. in Definition 1 the probability is taking over which variables ?  (Y given the X in the training set ?).
> >
> > Overall, I strongly believe that the paper introduces some interesting and novel ideas that deserve further investigation. However, I think the paper is far from being ready to be accepted. At least, the learnability of the partition function L(X) should be shown, which would facilitate  the understanding of the statistical problem under consideration. Also the authors will need to show why the problem at hand does not suffer from the intrinsic difficulties mentioned  by Vapnik & Izmailov. I hope these comments will help the authors to improve the quality of this work.

---

> > > ### Author Response · Authors · 2021-08-26
> > > **Clarifying the misunderstanding re our results**
> > >
> > > Thank you for your reply. Respectfully, we believe the reviewer has misunderstood the main result & motivations of our work. We try to clarify this below (please let us know if any concerns remain).
> > >
> > > The reviewer's summary of our work (the "statistical problem under consideration") is not accurate: we _do not_ introduce a new statistical problem to be solved. Rather, we study the behaviors of *existing* methods that perform classification with iid samples. In some sense, we are asking "what problem do these existing methods solve?". In particular, we empirically characterize the joint distribution $(x, f(x))$ learnt by current classification methods, and show that some statistical tests on this distribution are _similar_ to the true distribution $(x, y)$
> > >
> > > Thus, we do not try to directly "learn a partition function L", as the reviewer comments. We merely use partition functions as an analytical tool, to better characterize the similarity of the distributions of interest. In short, as the reviewer also alludes to, we claim that $(L(x),f(x))$ and $(L(x),y)$ are $\epsilon$-close in TV-distance for many partition functions $L$, or what we call Distinguishable features (formally defined in Definition 1).
> > >
> > > In a given setting (of architecture, training method and number of samples), there exist *many* partition functions for which distributional generalization holds (as specified by our Feature Calibration Conjecture and Definition 1).
> > > Here is an illustrative example: Consider the constant function L(x) = 0. That is, the partition into 1 part.
> > > This function is always a distinguishable feature, for *any* learning procedure, since it is trivially learnable from samples (under very weak assumptions).
> > > Even in this simple case, our conjecture implies a nontrivial behavior.
> > > It implies that: interpolating models have the correct *marginal distribution* on labels. That is, even "bad" models, with high test error, will have close to the correct *class-balance* of their outputs. This is a highly nontrivial and previously unformalized behavior.
> > > (This example also appears in L246 of the paper).
> > >
> > > Finally, the "questions of theoretical nature" posed by the reviewer are all addressed in our paper. We briefly describe the answers below:
> > > --- "Is the partition function L learnable from samples?": Yes, by definition. This definition is crucial to our work (see Definition 1, of Distinguishable Features). In our experimental validation for the Feature Calibration conjecture, we do not prove learnability for the partition functions, instead we show it empirically. For example: In all our experiments with the original labels of CIFAR-10 as the partition, a WideResNet learns it to accuracy >95%, hence it is learnable empirically.
> > > --- "How large M can be?": There are no a priori restrictions. However, if M is too large, then it may fail to be a Distinguishable Feature in a given setting of samples (according to Definition 1).
> > > --- "Can M be increased with the size of the training set n?": Yes. Note that Definition 1 depends on the training procedure $A$, the Distribution $D$, and the number of samples $n$).
> > > --- "Is there any connexion with the complexity of the class of models of the classifier?": Yes, a deep connection. Note that Definition 1 depends on the training procedure $A$.
> > >
> > > Further, we do not believe the objections posed by Vapnik & Izmailov apply in our setting, as we previously described. They are studying a fundamentally different problem.
> > >
> > > Regarding mathematical formalism: In Definition 1, the probability is taken over all the variables in the subscript: Random samples (x_i, L(x_i)) which form the training set S, randomness of the training procedure $A$, and randomness in the choice of test samples $x \sim D$.
> > > Any perceived mathematical imprecision is merely a notational issue (which we are happy to resolve).
> > > Finally, could the reviewer please clarify what they mean by "non-conclusive numerical validation", in light of our extensive quantitative experiments?

---

### Official Review · Reviewer_EwQL · 2021-07-21

**Rating:** 7
**Confidence:** 4

**Summary:**

This submission proposes a new type of generalization satisfied by interpolating classifiers (the primary focus), that is different from agreement between test error and train error as in classical learning theory. This type of generalization is largely motivated by empirical evidence across different models (neural networks, kernel regression & svm, decision trees) and theoretically for 1-Nearest Neighbors classification (Theorem 1). More specifically, the submission claims that for certain types of mappings x -> L(x) (termed "distinguishable features" -- i.e. they must be learnable to good test accuracy in the setting under consideration), the joint distribution of (L(x), y) where (x,y) pairs are drawn from the true distribution will be "close" to the joint distribution of (L(x), f(x)) on the test set, where f(x) is the trained model prediction.

List of contributions:
--Proposes that interpolating classifiers satisfy "distributional generalization." This is first more specifically formulated as the "Feature Calibration Conjecture" (FCC) in Section 3. A crucial component of this is to identify the types of features L(x) for which the calibration holds (Definition 1); roughly, if the feature is learnable from inputs x to good test accuracy in the setting under consideration (fixed algorithm, data distribution, number of samples, allowable error), then under that same setting calibration will be achieved. The authors prove FCC is true for 1-Nearest neighbors. The Feature Calibration Conjecture is not fully specified, in that there is an assumption of "natural distribution" D (which is such that the conjecture holds in the empirical settings considered).

--Empirical test of the FCC. The types of distinguishable features L considered include (i) the constant partition (calibration then implies the marginal distribution p(y) of the predicted labels matches that of the true label distribution); (ii) a "coarse partition" (L(x) is an "easy" classification task) and calibration holds for fine-grained label subgroups, for instance in Imagenet; (iii) "class partition" (classes that can be learned well from standard datasets, and the true label distribution is a noisy corruption of this); (iv) multiple features. Models investigated include neural networks (WideResnets and MLPs), kernels (standard and high-performing ones), and decision trees.

--A definition for "distributional generalization" which subsumes the definition of feature calibration & classical generalization, and very preliminary evidence that this holds for non-interpolating models (a Wide Resnet model on CIFAR-10 during training).


**Ethical Concerns:**

There are no ethical issues with the paper as far as I can tell.

**Limitations And Societal Impact:**

The paper does not address potential negative societal impact, but I believe it is not necessary given the nature of the work.

The authors do provide adequate discussion of limitations and are clear about the conjectural portions of their work.

**Main Review:**

Originality: This work identifies & investigates an original empirical property that appears to be satisfied by interpolating classifiers. The proposed concepts (Feature Calibration) are general & seem distinct from any prior work to the best of my knowledge. (For example, the authors note that "standard" calibration is a particular case of Feature Calibration with specific choice of L(x).) I think this work is bold in what it seeks to accomplish (asking a fundamentally new question at the basis of learning theory, building evidence from diverse empirical examples) and makes a reasonable first step towards doing so.

Quality: The quality of the work seems high. The authors give a number of different empirical examples with sufficient coverage across model diversity and provide a theoretically satisfied case (1-Nearest Neighbors). The authors note the limitations of the work (e.g. the FCC is specified in terms of "natural distributions" which has an unclear meaning) and are fairly thorough in addressing potential concerns & connections to other works. I was not able to identify significant shortcomings with respect to the conceptual formulation or message of the paper. I make some comments about diversity of empirical examples (with respect to datasets / partitions) below.

Clarity: The paper is clearly written and was straightforward to read.

Significance: I think this paper takes a risk in its premise, provides a reasonable amount of evidence in support of its main claims, and would spur further work in this area; as I was also not able to identify any major issues with the paper, this is the reason behind my suggested score. In Section 1.3 the authors discuss a number of consequences for learning theory which seem notable. This work tells us more about the implicit bias of models (though not just neural networks); most interesting I think is that in certain limits (e.g. when models are interpolating, and model size >> dataset size), models are not approaching the Bayes optimal classifier but sample from the distribution p(y|x).

Other comments & questions:

--I find some mentions of "Bayes optimal classifier" to be confusing at times -- for instance, its usage in L268 seems to be with respect to argmax p(y | L(x)) (rather than argmax p(y|x)), if I understand correctly. Perhaps it is worth pointing this out and the connection between the two (e.g. in the setting where p(y|x) is discussed, section starting at L107).

--In the empirical examples with label noise, why do the authors consider only sparse confusion matrices?

--I think this work has good empirical diversity across very distinct classes of interpolating models, but one place where the paper could be noticeably strengthened is diversity of examples with respect to the datasets / partitions L(x) that are considered. Apart from the constant partition L(x) = 0, all the partitions seem to be largely based off of semantic distinctions that are already available in datasets. (Because of this, I also wondered whether the behavior is largely inherited from the dataset and from the existing semantic structure within it.) Are there more complex L(x), that are not semantically constructed, that can be considered? Separate but related to this, is it valid to consider L(x) that have some knowledge of the distribution D, or even a particular training set? For instance, L(x) that is constructed from trained intermediate layer embeddings in a neural network -- or would L(x) that has knowledge of the distribution D potentially break FCC?

(For instance, this type of L(x) / y constructed from true labels / label noise on existing label structure, e.g. CIFAR-10, is also used in the non-interpolating setting in Section 5.2, which makes the reader wonder what is actually happening in the interpolating setting. More generally, I found Section 5.2 to be in tension with the main message prior to that section, and perhaps too preliminary to be presentable.)

--I think one other experiment that would significantly strengthen the paper is one that tests the implications mentioned for scaling limits in L107 (e.g. measure feature calibration as models are scaled from the underparametered scaling regime to overparameterized.) This would give some more diverse evidence of places where feature calibration breaks down.

Nit points:
--Typo L263: spare -> "sparse"
--Typo L265: Figure label should be "2A"
--L749 in appendix is cut off

**Time Spent Reviewing:**

6

---

> ### Author Response · Authors · 2021-08-09
> **Response to Reviewer EwQL**
>
> Thank you for your detailed and insightful review, and for recognizing that our work is "bold in what it seeks to accomplish", and "would spur further work in this area". We address your specific comments and questions below (and we are happy to clarify further if any concerns remain).
>
> **Re. "Bayes Optimality" (L268)**: Thank you for pointing out this potential confusion. Throughout the paper, we use "Bayes optimal classifier" to refer to the true Bayes optimal function, i.e. $f^*(x) := \textrm{argmax}_y p(y|x)$. However, in many cases this function "factorizes" through L(x), meaning $f^*$ is only a function of $L(x)$ and not directly of $x$ (eg: the sparse confusion matrix, where the noise depends only on the class of the example). This is likely the source of confusion, since in such cases we use p(y|x) and p(y|L(x)) interchangeably, since they are equivalent. We will clarify this notational issue in the revision.
>
> **Re. why only "sparse confusion matrices"**: This was just a choice made for presentation reasons, not scientific ones. The conjecture holds for arbitrary confusion matrices (including dense ones). We chose to use sparse matrices in the Figure so that the reader could quickly visually compare the structure of matrices (e.g. in Fig 2A), instead of having to compare many numbers in a dense matrix.
>
> **Re. "Are there more complex L(x), that are not semantically constructed, that can be considered?"**: This is a very interesting question -- the short answer is yes, we did several such experiments in the course of this research (which we describe below), but chose not to include them in favor of focusing on other results. We are happy to include these results in a revision. The long answer is below:
>
> There are two conceptually different cases. In the first case, L(x) is statistically independent of the labels y. In this case, the conclusion of the conjecture is fairly trivial: it implies that the outputs f(x) are *also* independent of L(x), and the outputs f(x) have the correct marginal distribution. This held empirically in our experiments, but we considered it a less-interesting behavior (though it has a certain implication for fairness, which is described in the response to Reviewer fCRG)
>
> In the second case, L(x) is correlated with the labels y. In this case, the feature L in some sense already contains some "semantic" content (since it is correlated with the labels). However, as you mention, it is still interesting to consider L(x) which are not directly determined from dataset labels.
>
> In both cases, we experimentally observed our conjecture to hold for the following choices of distinguishable feature L:
> 1. $L \in \\{0, 1\\}$  determined by a linear classifier on the pixel domain (we considered both random linear classifiers, which were independent of Y, and linear classifiers which were correlated with Y, obtained by training on the distribution).
> 2. $L \in \\{0, 1\\}$ determined by a "small" CNN (which was small enough to be learnable, i.e. distinguishable). We considered both random CNNs and pretrained CNNs.
>
> Finally, another family of distinguishable but non-semantic features L are those obtained by "combining" other features in some "easy-to-learn" way. For example, taking the XOR of a few binary features.
>
> **Re. "is it valid to consider L(x) that have some knowledge of the distribution D"**:
> Yes, as in above examples, it is valid for L(x) to be determined by some property of D (including properties that require training on D to define, e.g. the student-teacher setting).
> However, our conjectures do NOT allow L to have knowledge of the exact train set of f. Our current formalism does not allow this type of dependency to even be defined. However, it may be possible to extend our formalism to allow for this, although it is not immediately clear how to do so. This could be an interesting direction for future work.
>
> **Re. Section 5.2 may be "too preliminary"**: We agree that Section 5.2 is preliminary, and we openly acknowledge this in the paper as well. We included it because we believe there is still value to discussing results which we do not fully understand, in the hopes that they suggest a path to even deeper results, and inspire future work in the area. However, we appreciate the suggestion that this section may be more confusing than inspiring, and will consider this in the revision.
>
> **Re. testing the implications mentioned for scaling limits (L107)**: Thank you for this suggestion. We note that, since the submission of the current paper, there has been a new theoretical development (not by us) which proves the underparameterized scaling limit behavior that we mention: [https://arxiv.org/abs/2106.05932]. However, we agree that explicitly seeing the transition between regimes (e.g. by increasing the number of samples) would be an insightful experiment.
>
> There is also an experiment in the current submission which is very relevant: Figure 13 in the Appendix can be interpreted as moving from over-parameterized to under-parameterized regime by increasing the amount of L2 regularization. As seen in Figure 13, for high values of regularization (C=0.1) the MNIST classifier is close to Bayes-optimal, while for low values of regularization (C=100), the classifier is interpolating and not Bayes optimal (since we have introduced label noise, and Feature Calibration holds).

---

> > ### Comment · Reviewer_EwQL · 2021-09-01
> > **thanks for your response**
> >
> > Thanks to the authors for their response. I read through it and have no further questions.

---

### Decision · Program_Chairs · 2021-09-27

**Decision:**

Reject

**Comment:**

This paper studies interpolating classifiers and introduces a conjecture about their distributional generalization. Reviewer EwQL is mainly positive about this work. However, reviewers fCRG and fE8g, while appreciating interesting observations in the paper, find it not ready for publication.Specifically, within an internal discussion among the reviewers, fCRG stated how the conjecture introduced in the paper is meaningful or useful is not clear, saying "after reading the authors' response to my own review, I am now slightly more pessimistic about the possibility that distinguishability and feature calibration are tightly connected concepts". This reviewer expects the conjecture to be very persuasive and impactful for the paper to be accepted. Reviewer fE8g has concerns about the conjecture relying on poorly defined statistical setting and also recommends a more thorough discussion on the relationship between this work and that of Vapnik & Izmailov's. Authors are encouraged to address these concerns and resubmit.